# Are Language Models Efficient Reasoners?
# A Perspective from Logic Programming

**Andreas Opedal**[α,β,*]     **Yanick Zengaffinen**[α]     **Haruki Shirakami**[γ,δ]     **Clemente Pasti**[α]
**Mrinmaya Sachan**[α]     **Abulhair Saparov**[ε]     **Ryan Cotterell**[α]     **Bernhard Schölkopf**[α,β]

[α]ETH Zürich     [β]MPI for Intelligent Systems, Tübingen
[γ]EPFL     [δ]Idiap Research Institute     [ε]Purdue University

## Abstract

Modern language models (LMs) exhibit strong deductive reasoning capabilities, yet standard evaluations emphasize correctness while overlooking a key aspect of reasoning: *efficiency*. In real-world reasoning scenarios, much of the available information is irrelevant, and effective deductive inference requires identifying and ignoring such distractions. We propose a framework for assessing LM reasoning efficiency through the lens of logic programming, introducing a simple method to align proofs written in natural language—as generated by an LM—with shortest proofs found by executing the logic program. Efficiency is quantified by measuring how well a model avoids unnecessary inference. Empirically, we construct a dataset of math word problems injected with various number of irrelevant axioms that vary in semantic overlap with the goal theorem. We find that current LMs show marked accuracy declines under such conditions—even with minimal, domain-consistent distractions—and the proofs they generate frequently exhibit detours through irrelevant inferences.[2]

## 1 Introduction

Large language models (LMs) appear capable of solving a wide range of tasks that rely on deductive reasoning, particularly when post-trained with reinforcement learning (Lightman et al., 2024; DeepSeek-AI, 2025) and scaled to use more compute at test time (Wang et al., 2024; Muennighoff et al., 2025; Snell et al., 2025). However, emerging findings suggest recent reasoning models often generate more tokens than necessary to solve problems, even for simple deductive tasks (Chen et al., 2025; Pu et al., 2025). Such findings point towards a key dimension of deductive reasoning that standard evaluations of LMs' reasoning abilities fail to systematically assess—*efficiency*. Indeed, more abstractly, in most real-world reasoning tasks, more information is available than is necessary to solve the problem. This spurious information is not random: it often interacts with relevant information, enabling the derivation of true but irrelevant conclusions. Crucially, it is unknown *a priori* which pieces of information will be relevant for determining whether a desired conclusion is supported by the evidence. An *efficient* solution to a problem uses only necessary information and takes as few steps as possible.

To characterize efficiency, we adopt a formalization of deductive reasoning based on *logic programming* (Kowalski, 1974). Our perspective is that logic programming provides a clean and flexible framework for reasoning within a well-understood proof system. Given a logic program—that is, a set of inference rules and axioms—a proof of some goal theorem can be viewed as a path in a hypergraph induced by the inference rules, starting from vertices corresponding to axioms and terminating at a vertex corresponding to the goal theorem. Then, the *most efficient* proof is simply a shortest such path. Using this machinery, the goal of this paper is to evaluate a *language model's* reasoning efficiency. Thus, we require an additional mechanism to bridge the gap between reasoning

---

[*]Please direct correspondence to andreas.opedal@inf.ethz.ch, ryan.cotterell@inf.ethz.ch, and asaparov@purdue.edu.

[2]Code for this project is available at https://github.com/rycolab/reasoning-efficiency.

39th Conference on Neural Information Processing Systems (NeurIPS 2025).

in logic programming and reasoning in natural language. To this end, we introduce the notion of a *verbalized* logic program, in which each theorem in the logic program is associated with a set of natural language strings.[3] Verbalized logic programs allow us to map the deductions performed by an LM—expressed as a natural language proof—onto deductions performed during the execution of a logic program. While numerous recent papers use number of generated tokens as a proxy for efficiency (Arora & Zanette, 2025; Han et al., 2025; Ma et al., 2025), doing so conflates inefficiency stemming from two separate sources: (1) unnecessary deduction steps and (2) verbosity in the natural language strings expressing those deduction steps. In contrast, our framework disentangles these two factors, with our paper's experiments emphasizing the former.

Empirically, we construct verbalized logic programs for grade school math word problems (GSM problems; Cobbe et al., 2021; Li et al., 2024; Zhang et al., 2024), adopting methods from Opedal et al. (2025). Our primary experimental manipulative is the injection of irrelevant axioms into these GSM programs, which yields many possible implications that are irrelevant to a goal theorem of interest. We experiment on problems that vary both in how much the information in the irrelevant axioms *overlap* with the goal theorem, as well as in *how many* such axioms are injected, generalizing existing datasets that only include a single irrelevant axiom (Shi et al., 2023; Mirzadeh et al., 2025). We first measure accuracy, showing that current LMs are less accurate on problems containing irrelevant axioms than on equivalent problems without them. This performance gap often persists even in the simplest cases, where a single irrelevant axiom from the same domain is introduced, and grows larger as more irrelevant axioms are added. We confirm that this reduction in accuracy is not due to longer inputs alone: LMs usually perform better on control problems of equal length but without irrelevant content.

Next, we map the reasoning performed by the LM onto theorems they correspond to when executing the logic program. We find that while the LMs predict most of the correct intermediate theorems for the problems where they correctly generate the goal, they are often inefficient. In particular, for GSM problems where about half of the axioms are irrelevant, more than half of the LM's predicted theorems are irrelevant too, i.e., not needed for proving the goal. The LMs are particularly inefficient when the irrelevant axioms overlap semantically with the query—for instance, when the question asks *"how many cats does Ryan have?"* and the irrelevant axioms also mention *"Ryan"* or *"cats"*. On the other hand, these results also suggest that the LMs' search procedure sometimes employ a useful heuristic based on such overlap. Our results shed some light on how LMs, albeit in natural language, perform inference.

**Outline.** The remainder of the paper is structured as follows: §2 situates our contributions among related work. §3 provides relevant background on logic programming and discusses how reasoning efficiency is measured relative to shortest proofs. §4 introduces verbalized logic programs and the specifics of our GSM programs. §5 presents experiments and results on how LMs reason on verbalized GSM programs with irrelevant axioms. Apps. A-D give further technical details and empirical results.

## 2    Related Work

**Evaluating Reasoning.**    Most studies and benchmarks on LM reasoning evaluate correctness based on the LM's final answer (Hendrycks et al., 2021; Patel et al., 2021; Rein et al., 2024; Yu et al., 2024, *inter alia*). However, correctness of the final answer does not guarantee correctness of the proof (Lyu et al., 2023; Turpin et al., 2023). Some studies include more fine-grained reasoning evaluations by verifying LM-generated proofs (Gontier et al., 2020; Frieder et al., 2023; Nguyen et al., 2024; Wang et al., 2024; Petrov et al., 2025). While useful, many such evaluations rely either on manual scrutiny or heuristic measures of the proof's correctness. An alternative approach is to use proof assistants, e.g., Lean (de Moura & Ullrich, 2021), for formal verification (First et al., 2023; Tsoukalas et al., 2024); however, LMs may have been trained on less amounts of such data as compared to natural language. We perform an automatic evaluation by parsing the LM-generated output into proofs in logic programs.

**Irrelevant Information in Reasoning Tasks.**    Our work relates to papers that evaluate LMs' ability to correctly solve problems with irrelevant information (or missing information; Li et al., 2025). Shi et al. (2023) create such a dataset by appending a single irrelevant statement to problems taken from GSM8k (Cobbe et al., 2021). Mirzadeh et al. (2025) seem to take a similar, albeit more manual approach; however, details presented are scarce and their dataset has not yet been made publicly available. Xu et al. (2025) incorporate irrelevant statements as part of a pipeline for generating problem variations. Anantheswaran et al. (2025) use a prompting-based method for augmenting problems with sev-

---

[3]Previous work have made implicit use of similar notions (e.g., Betz et al., 2021; Clark et al., 2021; Saparov & He, 2023; Morishita et al., 2023; Han et al., 2024).

eral irrelevant statements. We formalize the notion of irrelevance through logic programming and generalize previous approaches by generating problems that may have several, arbitrarily placed irrelevant axioms, which can be used together in further inference. Thus, there are many implications that are irrelevant to the goal theorem and the challenge becomes not only to generate *correct* proofs, but proofs that only contain steps that are *necessary*. By generating new problems from scratch, our approach avoids bias from memorizing the efficient solution seen during training (see, e.g., Zhang et al., 2024).

**Search and Efficiency.** Other studies have investigated whether and how transformers can learn search tasks (Gandhi et al., 2023, 2024; Kazemi et al., 2023; Lehnert et al., 2024; Sanford et al., 2024; Sel et al., 2024; Shah et al., 2024; Saparov et al., 2025). We are interested not only in *whether* a transformer-based LM can perform accurate search, but in how *efficient* it is. Efficiency of large (reasoning-based) LMs is a rapidly growing area of research (Sui et al., 2025), due to their often lavish use of compute (Chen et al., 2025; Pu et al., 2025). Several methods have been proposed to make reasoning more efficient (Han et al., 2025; Ma et al., 2025), e.g., by incorporating length rewards in training (Arora & Zanette, 2025; Luo et al., 2025; Team et al., 2025). While these papers focus solely on the number of tokens, we argue that it is more informative to measure efficiency based on the natural language proof, since a long output can be explained either by unnecessary inference steps or by "verbose" verbalizations of the proof. Moreover, an LM should avoid generating redundant tokens (Xia et al., 2025), but it should also not skip necessary inference steps in favor of a shorter output.

# 3 Logic Programming and Deductive Reasoning

This section provides relevant background on logic programming (Kowalski, 1974). It also introduces the metric we propose for scoring a proof's efficiency in §3.2.

## 3.1 Typed Logic Programs with Built-ins

**Basic Notions.** A **signature** is a 3-tuple $\Sigma = (\Sigma_p, \Sigma_x, \Sigma_I)$, where $\Sigma_p$ is a set of **predicate** (or **relation**) symbols, denoted $p, q, r, ...$; $\Sigma_x$ is a set of constants, denoted $x, y, z, ...$; $\Sigma_I$ is a set of variables, denoted $X, Y, Z, ...$[4] Every predicate is associated with an arity, which we denote using the function **arity** $ar \colon \Sigma_p \to \mathbb{N}$, specifying how many arguments it takes. Arguments to predicates are called **terms**; they can be either constants (**ground** terms) or variables (**non-ground** terms). An **atomic formula**, called an **atom**, for short, is an expression of the form $p(t_1, ..., t_N)$, where $p \in \Sigma_p$ is a predicate symbol of arity $ar(p) = N$ and $t_1, ..., t_N \in \Sigma_x \cup \Sigma_I$ are all terms. The **Herbrand base** $H$ for signature $\Sigma$ is the set of all atoms that can be formed by terms in $\Sigma_x$, i.e., $H = \{p(t_1, ..., t_N) \mid p \in \Sigma_p, ar(p) = N, t_1, ..., t_N \in \Sigma_x\}$.[5] Subsets of the Herbrand base $I \subseteq H$ are called **interpretations**.

**Logic Programming.** An **inference rule** is an expression of the form $b_1, ..., b_N \vdash h$, where $b_1, ..., b_K, h$ are atoms; $b_1, ..., b_K$ is the **body** (or **premises**) and $h$ is the **head** (or **conclusion**). For example,

$$\mathrm{parent}(X, Y), \mathrm{ancestor}(Y, Z) \vdash \mathrm{ancestor}(X, Z)$$

is an inference rule that allows us to conclude that if $X$ is a parent of $Y$ and $Y$ is an ancestor of $Z$, then $X$ is an ancestor of $Z$. An inference rule is called **range restricted** if each variable appearing in the conclusion $h$ also appears in at least one atom $b_k$ in the premise. For example, $p(X) \vdash q(X)$ is range restricted, while $p(X) \vdash q(X, Y)$ is not. In this paper, we require all inference rules to be range restricted.[6] Inference rules with a null premise, i.e., where $K = 0$, and a ground conclusion, i.e., where $h \in H$, are called **axioms**. A set of axioms is denoted $A$. For example, $\vdash \mathrm{parent}(abraham, isaac)$, also written $\mathrm{parent}(abraham, isaac)$, omitting the $\vdash$ symbol, is an axiom. A **logic program** $\mathcal{P}$ over a signature $\Sigma$ is a set of inference rules in which all atoms are formed by symbols in $\Sigma$. The following is an example logic program, adapted from Sterling & Shapiro (1994, §5):

$$\mathrm{parent}(terah, abraham) \quad \mathrm{parent}(abraham, ishmael) \quad \mathrm{parent}(abraham, isaac)$$

$$\mathrm{parent}(X, Y) \vdash \mathrm{ancestor}(X, Y) \quad \mathrm{parent}(X, Y), \mathrm{ancestor}(Y, Z) \vdash \mathrm{ancestor}(X, Z)$$

---

[4]Many logic programming languages, e.g., Prolog (Colmerauer & Roussel, 1993; Körner et al., 2022), additionally have the notion of a function. Constants are then just nullary functions. Our notion of logic programming is most similar to Datalog (Vardi, 1982; Maier et al., 1984; Ceri et al., 1989), which does not.

[5]In the case that the signature additionally contains a set of function symbols $\Sigma_f$, the Herbrand base is defined as the set of all atoms that can be formed by all terms in the **Herbrand universe**, which is the smallest set $U$ that satisfies the equation $U = \Sigma_x \cup \{f(t_1, ..., t_N) \mid f \in \Sigma_f, ar(f) = N, t_1, ..., t_N \in U\}$.

[6]Range restriction ensures that applying the fixpoint operator (Eq. (1)) does not create non-ground atoms.

**Types and Built-ins.** Our notion of logic programming additionally includes types (Abiteboul et al., 1995, §21) and built-ins (Kaminski et al., 2017), which we define here. We partition $\Sigma_x$ and $\Sigma_\mathcal{X}$ into $T$ disjoint subsets, i.e., $\Sigma_x = \Sigma_x^1 \sqcup \cdots \sqcup \Sigma_x^T$ and $\Sigma_\mathcal{X} = \Sigma_\mathcal{X}^1 \sqcup \cdots \sqcup \Sigma_\mathcal{X}^T$, respectively, and associate each subset with a **type**. These subsets are paired index-wise, i.e., $(\Sigma_x^1, \Sigma_\mathcal{X}^1), ..., (\Sigma_x^T, \Sigma_\mathcal{X}^T)$, ensuring that the constant and variable types match. In this paper, we consider three types: (i) natural numbers, denoted $(\mathbb{N}_x, \mathbb{N}_\mathcal{X})$, (ii) strings, $(\Delta_x^*, \Delta_\mathcal{X}^*)$, and (iii) sets of strings, $(2_x^{\Delta^*}, 2_\mathcal{X}^{\Delta^*})$. Our introduction of types is necessitated by our desire to add additional power to our notion of logic programming that is external to the language itself. Specifically, we will introduce **built-in predicates**, simply called **built-ins** through the exposition, that add various arithmetic and set-theoretic operations. We enumerate these operations:

| | | | |
|---|---|---|---|
| $X_\mathbb{Z} + Y_\mathbb{Z} = Z_\mathbb{Z}$ | (Integer Addition), | $X_{2^{\Delta^*}} \cup Y_{2^{\Delta^*}} = Z_{2^{\Delta^*}}$ | (Set Union), |
| $X_\mathbb{Z} - Y_\mathbb{Z} = Z_\mathbb{Z}$ | (Integer Subtraction), | $X_{2^{\Delta^*}} \cap Y_{2^{\Delta^*}} = Z_{2^{\Delta^*}}$ | (Set Intersection), |
| $X_\mathbb{Z} \times Y_\mathbb{Z} = Z_\mathbb{Z}$ | (Integer Multiplication), | $|X_{2^{\Delta^*}}| = X_\mathbb{Z}$ | (Set Cardinality), |
| $X_\mathbb{Z} = Y_\mathbb{Z}$ | (Integer Equality), | $X_{2^{\Delta^*}} = Y_{2^{\Delta^*}}$ | (Set Equality), |
| $X_\mathbb{Z} \geq Y_\mathbb{Z}$ | (Integer Comparison), | $|X_{2^{\Delta^*}}| \geq X_\mathbb{Z}$ | (Set Cardinality Comparison). |

The truth value of grounded atoms constructed from built-in predicates is evaluated *externally* to the logic program. To do so, we define the **built-in evaluator** $\mathsf{eval} \colon H \to \{\mathtt{T}, \mathtt{F}\}$, that maps all ground built-ins that evaluate to true to $\mathtt{T}$ and all ground built-ins that evaluate to false to $\mathtt{F}$. Additionally, any element of $H$ that is *not* a built-in evaluates to $\mathtt{F}$. For example, $\mathsf{eval}(5+4=9) = \mathtt{T}$, $\mathsf{eval}(5+4=10) = \mathtt{F}$, and $\mathsf{eval}(\mathrm{parent}(abraham, isaac)) = \mathtt{F}$. For example, to illustrate the use of built-ins in a logic program, we can extend the inference rules from the earlier example to measure the depth of the ancestor relation (e.g., parent, grandparent, great-grandparent, etc):

$$\mathrm{parent}(X_{\Delta^*}, Y_{\Delta^*}) \vdash \mathrm{ancestor}(X_{\Delta^*}, Y_{\Delta^*}, 1)$$

$$\mathrm{parent}(X_{\Delta^*}, Y_{\Delta^*}), \ \mathrm{ancestor}(Y_{\Delta^*}, Z_{\Delta^*}, X_\mathbb{Z}), \ X_\mathbb{Z} + 1 = Y_\mathbb{Z} \vdash \mathrm{ancestor}(X_{\Delta^*}, Z_{\Delta^*}, Y_\mathbb{Z})$$

**Substitutions and Semantics.** To assign semantics to a logic program, we require a bit more machinery. A **substitution** $\theta$ is a finite set of pairs $\{(X_m, t_m)\}_{m=1}^M$, where $X_m \in \Sigma_\mathcal{X}$, $t_m \in \Sigma_x \cup \Sigma_\mathcal{X}$, $X_m \neq X_{m'}$ for all $m \neq m'$, and $X_m \neq t_m$ for all $m$ (Sterling & Shapiro, 1994, p. 14). Additionally, a **typed substitution** is a substitution where, if $X_m \in \Sigma_\mathcal{X}^k$, then $t_m \in \Sigma_x^k \cup \Sigma_\mathcal{X}^k$. We can apply a substitution $\theta$ to an atom $b$, denoted $b/\theta$, e.g., $\mathrm{parent}(X, isaac)/\{(X, abraham)\} = \mathrm{parent}(abraham, isaac)$. Let $\Theta(\mathcal{P})$ be the set of all typed substitutions under $\mathcal{P}$. We say that $b_1', ..., b_K' \vdash h'$ is an **instantiation** of $b_1, ..., b_K \vdash h$ if there exists a substitution $\theta \in \Theta(\mathcal{P})$ such that $b_1', ..., b_K' \vdash h' = b_1/\theta, ..., b_K/\theta \vdash h/\theta$. If an instantiation has no variables, we call it a **ground instantiation**. In addition, we say that an atom $b$ **unifies** with $b'$ if $\exists \theta \in \Theta(\mathcal{P}) : b/\theta = b'/\theta$. If $b$ unifies with $b'$ we write $b \equiv b'$. We define a logic program $\mathcal{P}$'s **fixpoint operator** $\mathrm{T}_\mathcal{P} \colon 2^H \to 2^H$ as

$$\mathrm{T}_\mathcal{P}(I) = I \cup (\{h/\theta \mid (b_1, ..., b_K \vdash h) \in \mathcal{P}, \ \wedge_{k=1}^K (b_k/\theta \in I \vee \mathsf{eval}(b_k/\theta)), \ \theta \in \Theta(\mathcal{P})\} \cap H), \ (1)$$

where $I \subseteq H$ is an interpretation. The fixpoint operator $\mathrm{T}_\mathcal{P}$ is **inflationary**, i.e., for every interpretation $I \subseteq H$, we have $b \in I \implies b \in \mathrm{T}_\mathcal{P}(I)$, and **monotone**, i.e., for every pair of interpretations $I_1, I_2 \subseteq H$, we have $I_1 \subseteq I_2 \implies \mathrm{T}_\mathcal{P}(I_1) \subseteq \mathrm{T}_\mathcal{P}(I_2)$. That $\mathrm{T}_\mathcal{P}$ is monotone allows us to employ least fixpoint semantics. To that end, we define the **minimal Herbrand model** as $M = \mathrm{T}_\mathcal{P}^*(A) \stackrel{\mathrm{def}}{=} \bigcup_{n=0}^\infty \mathrm{T}_\mathcal{P}^n(A)$, where $\mathrm{T}_\mathcal{P}^n$ denotes the $n$-fold application of the fixpoint operator $\mathrm{T}_\mathcal{P}$, i.e., $M$ is $\mathrm{T}_\mathcal{P}$'s least fixpoint.[7] Thus, $M$ is the subset of the Herbrand base that is true given the axioms and inference rules in the program; we call elements of $M$ **theorems**. Due to our inclusion of built-ins that encompass basic arithmetic operations, it is undecidable to compute $M$ (Dantsin et al., 2001), i.e., in general, we cannot decide whether $h \in \mathrm{T}_\mathcal{P}^*(A)$ for an arbitrary $h \in H$.

**Queries.** Given a logic program $\mathcal{P}$, we are often interested in determining whether there exists a theorem in $\mathcal{P}$ that is an instantiation of a specific (possibly non-ground) atom. We refer to such atoms as **queries**.[8] For example, building on the running example drawn from Sterling & Shapiro (1994, §5), we may wish to ask whether there exists a theorem that instantiates the atom $\mathrm{ancestor}(X, ishmael)$.

---

[7]Inflationarity is not needed to prove that the minimal Herbrand model $M$ exists. Indeed, monotonicity and the fact that interpretations of the Herbrand base form a complete lattice suffice to apply Tarski's (1955) theorem, which guarantees the existence of the least fixpoint. However, inflationarity does guarantee that, as we iteratively apply $\mathrm{T}_\mathcal{P}$, convergence to the least fixpoint is monotone.

[8]In principle, queries may also be ground; however, only the non-ground case is of theoretical interest here, as it extends beyond what can be handled by the machinery introduced so far.

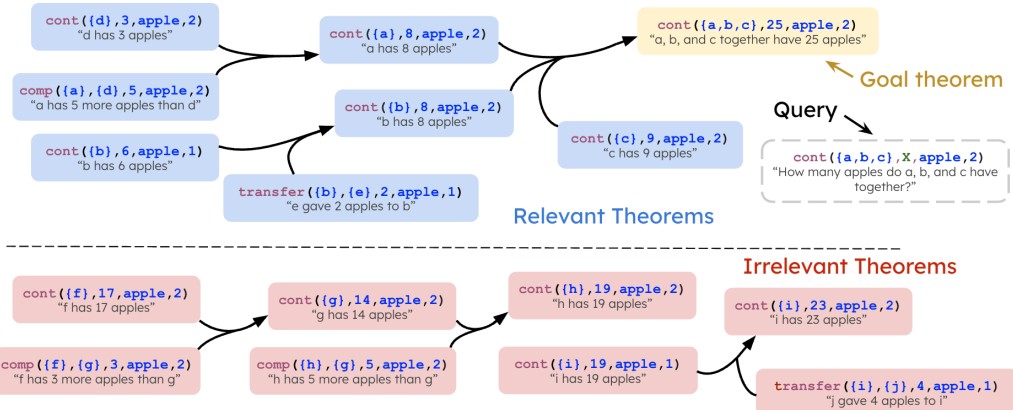

Figure 1: Example of a proof $\mathcal{P}$ of a **goal theorem** for the logic program presented in Table 1, with a shortest proof $\mathcal{P}^\star$ in **blue** and irrelevant theorems in **red**. We propose measuring efficiency as the size of the proof $|\mathcal{P}|$ relative to the size of the shortest proof $|\mathcal{P}^\star|$, penalizing the LM's proof for containing irrelevant theorems. We have omitted built-in predicates from this diagram for brevity.

Answering such a query amounts to finding all substitutions for $X$ that make the atom provable from $\mathcal{P}$. When $X$ can take on infinitely many instantiations, more sophisticated inference mechanisms—most notably, **unification** (Robinson, 1965)—are required to perform this kind of non-ground reasoning effectively. In this paper, however, we restrict attention to queries in which each variable is known *a priori* to range over a fixed, finite domain, which allows us to avoid additional complexities.

### 3.2 Deductive Reasoning

**Hypergraphs.** A **hypergraph** $\mathcal{G}$ (B-hypergraph; Gallo et al., 1993) is a tuple $(V, E)$, where $V$ is a set of vertices, and $E \subseteq 2^V \times V$ is a set of **hyperedges**, where a hyperedge $e = T \rightarrowtail v_h$ consists of a **tail** $T \subseteq V$, with $|T| > 0$, and a **head** $v_h \in V$. We define the size of a hypergraph as $|\mathcal{G}| \stackrel{\text{def}}{=} |V|$ where $\mathcal{G} = (V, E)$.[9] A **subhypergraph** of a hypergraph $\mathcal{G} = (V, E)$ is a hypergraph $\mathcal{G}' = (V', E')$ where $V' \subseteq V$ and $E' \subseteq E$. Given $S \subseteq V$ and $v \in V$, an $(S, v)$-**hyperpath** in a hypergraph $\mathcal{G} = (V, E)$ is a finite sequence of distinct hyperedges $T_1 \rightarrowtail v_{h_1}, ..., T_J \rightarrowtail v_{h_J}$ such that $v_{h_J} = v$ and for every $j \in [J]$: $T_j \subseteq S \cup \{v_{h_1}, ..., v_{h_{j-1}}\}$, i.e., each hyperedge's tail consists only of nodes that are either in the source set $S$ or are heads of previous hyperedges in the sequence. A hyperpath generalizes the notion of a directed path in a graph, but allows each hyperedge to have multiple tail nodes that jointly produce a head node. Finding the shortest $(S, v)$-hyperpath in a hypergraph is analogous to context-free parsing (Klein & Manning, 2001) and can be executed in polynomial time.

**Proof Forests, Proofs and Proof Efficiency.** Let $\mathcal{P}$ be a logic program. A **proof forest** $(\mathcal{F}, \ell)$ in $\mathcal{P}$ is a pair where $\mathcal{F} = (E, V)$ is a hypergraph and $\ell : V \to H$ where $H$ is $\mathcal{P}$'s Herbrand base (Heijltjes, 2010). Additionally, we require that, for every hyperedge $e = \{t_1, ..., t_K\} \rightarrowtail v_h \in E$, there exists a rule $(b_1, ..., b_K \vdash h) \in \mathcal{R}$ and a substitution $\theta \in \Theta(\mathcal{P})$ such that $b_1/\theta = \ell(t_1), ..., b_K/\theta = \ell(t_K)$ and $h/\theta = \ell(v_h)$. We call a proof forest $(\mathcal{F}, \ell)$ an $(A, h_g)$-**proof** if there exists a $(\ell^{-1}(A), \ell^{-1}(h_g))$-hyperpath in $(\mathcal{F}, \ell)$.[10] In Fig. 1, we show an example of a proof in which $h_g = \text{cont}(\{a,b,c\}, 25, apple, 2)$ in our custom logic program for math word problems (§4), together with some axioms in the program that do not contribute to the proof of the goal theorem $h_g$. We call an $(A, h_g)$-proof a **shortest proof** if it has the least number of vertices of all $(A, h_g)$-proofs in $\mathcal{P}$. A shortest proof can be found by forward-chaining, discussed in the subsequent paragraph. Now we turn to measuring proof efficiency. Consider an $(A, h_g)$-proof $\mathcal{P}$ in $\mathcal{P}$. We define the **efficiency** of $\mathcal{P}$ as $\text{EFFICIENCY}(\mathcal{P}) \stackrel{\text{def}}{=} |\mathcal{P}^\star|/|\mathcal{P}|$ where $|\mathcal{P}^\star|$ is the number of vertices in a shortest $(A, h_g)$-proof. In the remainder of the paper, we will refer to an axiom $a$ as **irrelevant** if there does not exist a shortest proof $\mathcal{P}$ that contains a vertex $v$ such that $\ell(v) = a$.

---

[9]We note that this is non-standard; the size of a hypergraph is more often defined as its number of hyperedges or the sum of the cardinalities of its hyperedges (Gallo et al., 1993). We use this definition to sync with the experimental setup, which we explain in §5. Future work could easily adapt our efficiency metric to other definitions.

[10]We note that the expression $(A, h_g)$-hyperpath is a slight abuse of notation in the case that $\ell$ is not injective adopted for convenience: $A$ and $h_g$ are a subset and an element, respectively, of $\mathcal{P}$'s Herbrand base—not of $\mathcal{R}_\mathcal{P}$'s $V$. In this context, by $A$, we refer to a set $X \subseteq \ell^{-1}(A)$ where $\ell(X) = A$, and by $h_g$ we refer to a set $Y \subseteq \ell^{-1}(h_g)$ where $\ell(Y) = h_g$.

| id | Inference Rule |
|---|---|
| (1a) | $\mathrm{cont}(A_{\mathcal{A}}, X_{\mathbb{N}}, E_{\Delta^*}, T_{\mathbb{N}}), \mathrm{comp}(B_{\mathcal{A}}, A_{\mathcal{A}}, Y_{\mathbb{N}}, E_{\Delta^*}, T_{\mathbb{N}}), X_{\mathbb{N}}+Y_{\mathbb{N}}=Z_{\mathbb{N}} \vdash \mathrm{cont}(B_{\mathcal{A}}, Z_{\mathbb{N}}, E_{\Delta^*}, T_{\mathbb{N}})$ |
| (1b) | $\mathrm{cont}(B_{\mathcal{A}}, Z_{\mathbb{N}}, E_{\Delta^*}, T_{\mathbb{N}}), \mathrm{comp}(B_{\mathcal{A}}, A_{\mathcal{A}}, Y_{\mathbb{N}}, E_{\Delta^*}, T_{\mathbb{N}}), Z_{\mathbb{N}} \geq Y_{\mathbb{N}}, Z_{\mathbb{N}}-Y_{\mathbb{N}}=X_{\mathbb{N}} \vdash \mathrm{cont}(A_{\mathcal{A}}, X_{\mathbb{N}}, E_{\Delta^*}, T_{\mathbb{N}})$ |
| (1c) | $\mathrm{cont}(A_{\mathcal{A}}, X_{\mathbb{N}}, E_{\Delta^*}, T_{\mathbb{N}}), \mathrm{cont}(B_{\mathcal{A}}, Y_{\mathbb{N}}, E_{\Delta^*}, T_{\mathbb{N}}), Y_{\mathbb{N}} \geq X_{\mathbb{N}}, Y_{\mathbb{N}}-X_{\mathbb{N}}=Z_{\mathbb{N}} \vdash \mathrm{comp}(B_{\mathcal{A}}, A_{\mathcal{A}}, Z_{\mathbb{N}}, E_{\Delta^*}, T_{\mathbb{N}})$ |
| (2a) | $\mathrm{cont}(A_{\mathcal{A}}, X_{\mathbb{N}}, E_{\Delta^*}, T_{\mathbb{N}}), \mathrm{transfer}(A_{\mathcal{A}}, B_{\mathcal{A}}, Y_{\mathbb{N}}, E_{\Delta^*}, T_{\mathbb{N}}), X_{\mathbb{N}}+Y_{\mathbb{N}}=Z_{\mathbb{N}}, T_{\mathbb{N}}+1=U_{\mathbb{N}} \vdash \mathrm{cont}(A_{\mathcal{A}}, Z_{\mathbb{N}}, E_{\Delta^*}, U_{\mathbb{N}})$ |
| (2b) | $\mathrm{cont}(B_{\mathcal{A}}, Z_{\mathbb{N}}, E_{\Delta^*}, T_{\mathbb{N}}), \mathrm{transfer}(A_{\mathcal{A}}, B_{\mathcal{A}}, Y_{\mathbb{N}}, E_{\Delta^*}, T_{\mathbb{N}}), Z_{\mathbb{N}} \geq Y_{\mathbb{N}}, Z_{\mathbb{N}}-Y_{\mathbb{N}}=X_{\mathbb{N}}, T_{\mathbb{N}}+1=U_{\mathbb{N}} \vdash \mathrm{cont}(B_{\mathcal{A}}, X_{\mathbb{N}}, E_{\Delta^*}, U_{\mathbb{N}})$ |
| (3) | $\mathrm{cont}(A_{\mathcal{A}_1}, Q_{\mathbb{N}_1}, E_{\Delta^*}, T_{\mathbb{N}}), ..., \mathrm{cont}(A_{\mathcal{A}_k}, Q_{\mathbb{N}_k}, E_{\Delta^*}, T_{\mathbb{N}}), A_{\mathcal{A}_1} \cup \cdots \cup A_{\mathcal{A}_k}=B_{\mathcal{A}}, Q_{\mathbb{N}_1}+\cdots+Q_{\mathbb{N}_k}=Y_{\mathbb{N}} \vdash \mathrm{cont}(B_{\mathcal{A}}, Y_{\mathbb{N}}, E_{\Delta^*}, T_{\mathbb{N}})$ 
 for $2 \leq k \leq K$ |
| (4) | $\mathrm{cont}(A_{\mathcal{A}}, X_{\mathbb{N}}, E_{\Delta^*}, T_{\mathbb{N}}), \mathrm{rate}(A_{\mathcal{A}}, Y_{\mathbb{N}}, E_{\Delta^*}, F_{\Delta^*}, T_{\mathbb{N}}), X_{\mathbb{N}} \times Y_{\mathbb{N}}=Z_{\mathbb{N}} \vdash \mathrm{cont}(A_{\mathcal{A}}, Z_{\mathbb{N}}, F_{\Delta^*}, T_{\mathbb{N}})$ |
| (5a) | $\mathrm{cont}(A_{\mathcal{A}}, X_{\mathbb{N}}, E_{\Delta^*}, T_{\mathbb{N}}), \mathrm{comp}(D_{\mathcal{A}}, C_{\mathcal{A}}, Y_{\mathbb{N}}, E_{\Delta^*}, T_{\mathbb{N}}), \mathrm{compeq}(A_{\mathcal{A}}, B_{\mathcal{A}}, C_{\mathcal{A}}, D_{\mathcal{A}}, E_{\Delta^*}, T_{\mathbb{N}}), X_{\mathbb{N}}+Y_{\mathbb{N}}=Z_{\mathbb{N}} \vdash \mathrm{cont}(B_{\mathcal{A}}, Z_{\mathbb{N}}, E_{\Delta^*}, T_{\mathbb{N}})$ |
| (5b) | $\mathrm{comp}(B_{\mathcal{A}}, A_{\mathcal{A}}, X_{\mathbb{N}}, E_{\Delta^*}, T_{\mathbb{N}}), \mathrm{comp}(D_{\mathcal{A}}, C_{\mathcal{A}}, Y_{\mathbb{N}}, E_{\Delta^*}, T_{\mathbb{N}}), X_{\mathbb{N}}=Y_{\mathbb{N}} \vdash \mathrm{compeq}(A_{\mathcal{A}}, B_{\mathcal{A}}, C_{\mathcal{A}}, D_{\mathcal{A}}, E_{\Delta^*}, T_{\mathbb{N}})$ |
| (6) | $\mathrm{p}(A_{\mathcal{A}}, E_{\Delta^*}, T_{\mathbb{N}}, ...), \neg\mathrm{transfer}(A_{\mathcal{A}}, B_{\mathcal{A}}, Y_{\mathbb{N}}, E_{\Delta^*}, T_{\mathbb{N}}), \neg\mathrm{transfer}(B_{\mathcal{A}}, A_{\mathcal{A}}, Z_{\mathbb{N}}, E_{\Delta^*}, T_{\mathbb{N}}), T_{\mathbb{N}}+1 = U_{\mathbb{N}} \vdash \mathrm{p}(A_{\mathcal{A}}, E_{\Delta^*}, U_{\mathbb{N}}, ...)$ 
 for $\mathrm{p} \in \{\mathrm{cont}, \mathrm{comp}, \mathrm{rate}, \mathrm{compeq}\}$ |

Table 1: Inference rules in $\mathcal{P}_W$. The symbols cont, comp, transfer, rate, compeq are predicates and we use variables $A_{\mathcal{A}}, B_{\mathcal{A}}, C_{\mathcal{A}}, C_{\mathcal{A}} \in 2_{\mathcal{I}}^A$ for agents, $E_{\Delta^*}, F_{\Delta^*} \in E_{\mathcal{I}}$ for entities, $X_{\mathbb{N}}, Y_{\mathbb{N}}, Z_{\mathbb{N}} \in \mathbb{N}_{\mathcal{I}}^q$ for quantities, and $T_{\mathbb{N}} \in \mathbb{N}_{\mathcal{I}}^t$ for timestamps. We refer to Fig. 2 for ground instantiations of the atoms shown here with corresponding example verbalizations. The symbol p in Rule (6) is a placeholder for any predicate other than transfer; because the predicates have different arities, we use "..." to denote other arguments that may be present. Rule (6) is special since it has negation (App. A.1); it is included to treat complications resulting from the timestamp in Rules (2a) and (2b)—see footnote 11—and is not used when generating shortest proofs for our experiments (App. C).

**Forward Chaining.** Forward chaining (Hayes-Roth et al., 1983; Poole & Mackworth, 2017) is a meta-strategy for theorem proving in logic programming that proceeds from the axioms toward the goal theorem. The process terminates once the goal theorem is proved. Pseudocode for the forward-chaining is given in Alg. 1 in App. A.2. As a meta-strategy, each instance of forward chaining defines an ordering over proof steps. Different orderings give rise to familiar search algorithms, such as depth-first search (DFS; Tarjan, 1972), breadth-first search (BFS; Moore, 1959), Dijkstra's (1959) algorithm, and heuristic-based search (Pearl, 1984) like A* (Hart et al., 1968). While DFS and BFS ignore information about the goal theorem, such information can guide search more efficiently, as exemplified by goal-aware strategies like Earley's (1970) algorithm for context-free parsing or its more general equivalent in logic programming—magic templates (Bancilhon et al., 1986; Ramakrishnan, 1991; Eisner & Blatz, 2007). In the spirit of using top-down information in proof search, in §5, we examine whether LMs make use of lexical overlap with the goal theorem as part of their internal search heuristic.

# 4 Evaluating Language Models on Grade School Math Word Problems

We are interested in reasoning that takes places in *natural language*, particularly as performed by LMs. To this end, we consider grade school math (GSM) word problems as empirical test domain. Such problems are commonly used for training and evaluating LMs on reasoning tasks (Cobbe et al., 2021; Patel et al., 2021). In §4.1, we introduce a family of logic programs, using the technical notions introduced in §3, that correspond to a natural class of such math word problems. We also describe a simple manner to convert text generated by an LM to a proof in such logic programs in §4.2. Finally, in §4.3 we explain how we generate problem descriptions in natural language that contain irrelevant axioms.

## 4.1 Modeling Math Word Problems with Verbalized Logic Programs

**Logic Programming for Grade School Math Word Problems.** We consider a particular family of logic programs to represent GSM problems, adapted from Opedal et al. (2023, 2025). All of the logic programs have the signature $\Sigma^W = (\Sigma_p^W, \Sigma_x^W, \Sigma_{\mathcal{I}}^W)$. The set of predicate symbols $\Sigma_p^W = \{\mathrm{cont}, \mathrm{comp}, \mathrm{transfer}, \mathrm{rate}, \mathrm{compeq}\}$ corresponds to arithmetic concepts that occur in GSM word problems (Riley et al., 1983), e.g., cont for denoting how many entities an agent contains, comp for comparing the number of entities across multiple agents, or transfer for expressing one agent transfering entities to another. We partition $\Sigma_x^W = 2_x^A \sqcup E_x \sqcup \mathbb{N}_x^q \sqcup \mathbb{N}_x^t$ and $\Sigma_{\mathcal{I}}^W = 2_{\mathcal{I}}^A \sqcup E_{\mathcal{I}} \sqcup \mathbb{N}_{\mathcal{I}}^q \sqcup \mathbb{N}_{\mathcal{I}}^t$ into the following pairs: sets of strings $(2_x^A, 2_{\mathcal{I}}^A)$ called **sets of agents** (*who* possesses), strings $(E_x, E_{\mathcal{I}})$ called **entities** (*what* is possessed), natural numbers $(\mathbb{N}_x^q, \mathbb{N}_{\mathcal{I}}^q)$ called **quantities** (*how much* is possessed), and natural numbers $(\mathbb{N}_x^t, \mathbb{N}_{\mathcal{I}}^t)$ called **timestamps** (*time* of possession). We note that $2_x^A$ is a power set of the set of agent strings $A_x$, which enables us to code a state where multiple agents possess the same entity jointly. The family of logic programs all share the same inference rules, which are given in Table 1. Fig. 1 shows an example proof using these inference rules, omitting built-ins. We note that possession may change over time and is governed by the transfer predicate in Rule (2).[11]

---

[11]The time component requires the addition of Rule (6), which expresses that all theorems with time $T_{\mathbb{N}}$ that are *not* affected by a proof step using Rule (2) maintain their truth value at $T_{\mathbb{N}}+1$. This is an instance of the

**Axioms.** The rules given in Table 1 are held constant across all logic programs in our family. Indeed, what distinguishes one program from another, then, is the choice of axioms. For example, consider the axiom cont(*{ryan}*, *5*, *cat*, *2*) which expresses that *"Ryan has 5 cats at time step 2"*; the predicate cont represents the semantics of possession, and *{ryan}* $\in 2_x^A$, *5* $\in \mathbb{N}_x^q$, *cat* $\in E_x$, and *2* $\in \mathbb{N}_x^t$ are all ground atoms of different types. Additionally, we only consider axiom sets $A_W \subset \overline{H}$ that have the following property: We call an interpretation $I \subseteq \overline{H}$ of a logic program in our family (Table 1) **numerically consistent** if there do *not* exist two substitutions $\theta = \{(A_{\mathcal{A}}, t_a), (X_{\mathbb{Z}}, t), (E_{\Delta^*}, t_e), (T_{\mathbb{N}}, t_t)\}$ and $\theta' = \{(A_{\mathcal{A}}, t_a), (X_{\mathbb{Z}}, t'), (E_{\Delta^*}, t_e), (T_{\mathbb{N}}, t_t)\}$ such that $t \neq t'$ and both cont($A_{\mathcal{A}}, X_{\mathbb{N}}, E_{\Delta^*}, T_{\mathbb{N}})/\theta$, cont($A_{\mathcal{A}}, X_{\mathbb{N}}, E_{\Delta^*}, T_{\mathbb{N}})/\theta' \in T^*_{\mathcal{P}_W}(I)$. This ensures that the minimal Herbrand model does not contain contradictory pairs such as cont(*{ryan}*, *5*, *cat*, *2*) and cont(*{ryan}*, *4*, *cat*, *2*). For example, the following set of axioms is numerically consistent: $A_W = \{$cont(*{ryan}*, *5*, *cat*, *2*), comp(*{eleanor}*, *{ryan}*, *3*, *cat*, *2*)$\}$, expressing that *"Ryan has 5 cats"* and that *"Eleanor has 3 more cats than Ryan"*. If we were to, e.g., include the two axioms $\{$cont(*{andreas}*, *7*, *cat*, *2*), comp(*{eleanor}*, *{andreas}*, *3*, *cat*, *2*)$\}$, we would obtain a numerically inconsistent axiom set. Inference in our family of logic programs is decidable under numerically consistent axiom sets. See App. B for details.

### 4.2 Language Modeling and Proofs in Natural Language

**Language Modeling.** We give a brief formal introduction to language modeling. Let $\Gamma$ be an alphabet of tokens and $\Gamma^*$ be the set of all strings over $\Gamma$, its Kleene closure. We write $\boldsymbol{w} \in \Gamma^*$ for a string, $w_t$ for the token at the $t^{\text{th}}$ position in $\boldsymbol{w} = w_1 \cdots w_T$, and $|\boldsymbol{w}| = T$ for the number of tokens in $\boldsymbol{w}$, i.e., its length. A **language model** (**LM**) $p$ is a probability distribution on $\Gamma^*$. Let EOS $\notin \Gamma$ be a distinguished symbol denoting the end of a token string. The probability of a string $\boldsymbol{w}$ can be written autoregressively as $p(\boldsymbol{w}) = \overrightarrow{p}(\text{EOS} \mid \boldsymbol{w}) \prod_{t=1}^{|\boldsymbol{w}|} \overrightarrow{p}(w_t \mid \boldsymbol{w}_{<t})$, where $\overrightarrow{p}(\cdot \mid \boldsymbol{c})$ is a probability distribution over $\overline{\Gamma} \overset{\text{def}}{=} \Gamma \cup \{\text{EOS}\}$ conditioned on the context $\boldsymbol{c} \in \Gamma^*$.

**Verbalized Logic Programs.** To bridge reasoning in formal proof systems to reasoning in natural language, we introduce the idea of a **verbalized logic program**. Given a logic program $\mathcal{P}$ with negation (App. A.1), let $\overline{H}$ be its extended Herbrand base. We associate each atom $b \in \overline{H}$ with a set of natural language strings. To that end, we define a **verbalizer** $\nu_{\mathcal{P}}: \overline{H} \to 2^{\Gamma^*}$, where each $\nu_{\mathcal{P}}(b)$ is a disjoint set. For every $b \in \overline{H}$, the set $\nu_{\mathcal{P}}(b)$ represents the various ways in which the meaning of $b$ can be expressed in natural language. In this paper, we take a straightforward approach. For each $b \in \overline{H}$, we

| Grounded Atom | Verbalization |
|---|---|
| cont({*alice*}, *3*, *apple*, *1*) | *"Alice has 3 apples."* |
| cont({*alice, bob*}, *8*, *apple*, *1*) | *"Alice and Bob have 8 apples combined."* |
| comp({*bob*}, {*alice*}, *2*, *apple*, *1*) | *"Bob has 2 more apples than Alice."* |
| transfer({*alice*}, {*bob*}, *2*, *apple*, *1*) | *"Bob gave 2 apples to Alice."* |
| rate({*alice*}, *4*, *basket*, *apple*, *1*) | *"Each of Alice's baskets contains 4 apples."* |
| *3+2=5* | Not verbalized |

Figure 2: Examples of verbalized grounded atoms, i.e., elements of $\nu_{\mathcal{P}_W}(b)$ for some $b \in H$. Built-ins are not verbalized other than as part of the conclusion; App. D.2 shows how this was done in our prompt. We additionally note that time is not verbalized directly but implicitly through the use of a past form depending on context.

construct a finite set $G_b \subset \Gamma^*$. Moreover, we enforce disjointness, i.e., $G_{b_2} \cap G_{b_1} = \varnothing$ for $b_1, b_2 \in \overline{H}$, and that, for all $b \in H$, each string in $G_b$ ends in a distinguished separator symbol—in our case a period "."—that appears nowhere else in the string. These assumptions allow for trivial linear-time parsing of natural language text into a sequence of atoms in the verbalized logic programs. In practice, we generate the strings in each $G_b$ with a series of hand-written templates. We give examples in Fig. 2.

### 4.3 Generating Problems with Irrelevant Axioms

**Sampling Axiom Sets.** We introduce a simple sampling algorithm, presented and analyzed in App. C, that samples a ground goal theorem $h_g \equiv$ cont($A_{\mathcal{A}}, X_{\mathbb{N}}, E_{\Delta^*}, T_{\mathbb{N}}$) and a shortest proof of $h_g$ under the rules $\mathcal{R}_W$ from Table 1. By construction, the axioms $A_W$ in the shortest proof are numerically consistent (§4.1) and *all* axioms in $A_W$ will be used in the proof of $h_g$ in the program $\mathcal{P}_W = \mathcal{R}_W \sqcup A_W$; we denote this dependency by $A_W(h_g)$.[12] That is, $A_W(h_g)$ contains no irrelevant

---

frame problem (McCarthy & Hayes, 1969; Sandewall, 1972; Hanks & McDermott, 1987). To express this rule, we introduce negation into the logic program; see App. A.1 for background on negation in logic programming. The only negated predicate in Table 1 is ¬transfer; because ¬transfer only appears in the body of an inference rule, the program is trivially stratified (App. A.1). That is ¬transfer atoms can not be proved true, but we assume ¬transfer atoms if the corresponding transfer atom is not known to be true from the axioms.

[12] $A_W(h_g)$ is the unique smallest set of axioms (Lemma 1) needed to prove $h_g$, justifying the function notation.

| Model | Base | w/ Axiom | | w/ Tree | | w/ Multiple Trees | |
|---|---|---|---|---|---|---|---|
| | | Irrelevant | Control | Irrelevant | Control | Irrelevant | Control |
| Llama-3.1-8B | 65.0 | 60.8 | 54.0 | 52.6 | 58.6 | 41.0 | 52.4 |
| | $(-4.3, 4.1)$ | $(-2.2, 2.1)$ | $(-4.4, 4.3)$ | $(-2.2, 2.2)$ | $(-4.4, 4.2)$ | $(-2.2, 2.2)$ | $(-4.4, 4.3)$ |
| Qwen2.5-Math-7B | 52.8 | 43.4 | 48.4 | 35.6 | 40.2 | 23.5 | 30.6 |
| | $(-4.4, 4.3)$ | $(-2.2, 2.2)$ | $(-4.4, 4.4)$ | $(-2.1, 2.2)$ | $(-4.2, 4.4)$ | $(-1.8, 1.9)$ | $(-3.9, 4.2)$ |
| QwQ-32B | 63.4 | 62.2 | 76.2 | 58.7 | 72.2 | 50.1 | 65.4 |
| | $(-4.3, 4.1)$ | $(-2.2, 2.1)$ | $(-3.9, 3.5)$ | $(-2.2, 2.1)$ | $(-4.1, 3.7)$ | $(-2.2, 2.2)$ | $(-4.3, 4.0)$ |
| DeepSeek-R1 | 99.4 | 98.2 | 98.4 | 98.0 | 98.6 | 97.6 | 98.6 |
| | $(-1.1, 0.4)$ | $(-0.7, 0.5)$ | $(-1.5, 0.8)$ | $(-1.1, 0.9)$ | $(-0.8, 0.6)$ | $(-1.1, 0.9)$ | $(-1.5, 0.7)$ |

Table 2: Average answer accuracy (%) with 95% confidence interval (CI) for base problems augmented with different degrees of irrelevance (one irrelevant axiom, one irrelevant tree, and multiple irrelevant trees; §4.3). The control problems are of the same length as the corresponding problems that have irrelevant axioms, with the difference being that all axioms are relevant. The CIs are Wilson (1927) score intervals. Note that adding irrelevant axioms degrades performance across all models.

axioms. However, to produce logic programs that *do* contain irrelevant axioms, we do the following. We sample an additional $M$ distinct distractor theorems $\{\widetilde{h}_m\}_{m=1}^M$. For each distractor theorem $\widetilde{h}_m$, we again apply our axiom sampling procedure to generate distractor axioms $\widetilde{A}_m(\widetilde{h}_m)$. Importantly, we are able to show that $A_{\mathrm{W}}(h_g)$ remain on a shortest proof of $h_g$ *even* in the case that we consider the augmented logic program $\widetilde{\mathcal{P}}_{\mathrm{W}} = \mathcal{R}_{\mathrm{W}} \sqcup A_{\mathrm{W}}(h_g) \sqcup \widetilde{A}_1(\widetilde{h}_1) \sqcup \cdots \sqcup \widetilde{A}_M(\widetilde{h}_M)$; see App. C.4.

**Structural Overlap.** In our experiments, we vary the size of the irrelevant axiom sets $\widetilde{A} = \widetilde{A}_1(\widetilde{h}_1) \sqcup \cdots \sqcup \widetilde{A}_M(\widetilde{h}_M)$ as follows: (i) a single irrelevant axiom (**w/ axiom**), where $M = 1$, $|\widetilde{A}_1(\widetilde{h}_1)| = 1$, and $\widetilde{A}_1(\widetilde{h}_1) = \{\widetilde{h}_1\}$ (i.e., the distractor theorem is trivial); (ii) a single irrelevant tree (**w/ tree**), where $M = 1$, $|\widetilde{A}_1(\widetilde{h}_1)| > 1$, and $\widetilde{h}_1$ has a non-trivial proof; (iii) multiple irrelevant trees (**w/ multiple trees**), where we set $M = 3$, $|\widetilde{A}_m(\widetilde{h}_m)| > 1$ for $m \in \{1, 2, 3\}$, and $\widetilde{h}_1, \widetilde{h}_2, \widetilde{h}_3$ all have non-trivial proofs.

**Agent and Entity Overlap.** We additionally control for overlap between the set of agents, i.e., elements of $2_x^A$, and the entities, i.e., elements of $E_x$, of the goal theorem $h_g \equiv \mathrm{cont}(A_{\mathcal{A}}, X_{\mathbb{N}}, E_{\Delta^*}, T_{\mathbb{N}})$, and those present in $\widetilde{A}$. Intuitively, for a query asking how many telescopes *"Bernhard"* has, we should make use of the information in *"telescope"* and *"Bernhard"* when deciding whether to take a certain deduction step. By constructing irrelevant axioms that also mention *"telescope"* and/or *"Bernhard"*, it becomes harder to distinguish what is relevant and what is not. We distinguish four cases: (i) neither the set of agents nor the entity occur in $\widetilde{A}$ (**no overlap**), (ii) the agent does not occur in $\widetilde{A}$ but the entity does (**entity overlap**), (iii) the entity does not occur in $\widetilde{A}$ but the set of agents does (**agent overlap**), and (iv) entity overlap in which the agents occurring in $\widetilde{A}$ have *lexical* overlap with the set of agents (**agent and entity overlap**), e.g., if `bernhard` is in $h_g$ then $\widetilde{A}$ contains agents like `bernhard's_student` or `bernhard's_son`. Entities that do *not* overlap are always made topically related, e.g., if $A_{\mathrm{W}}$ contains axioms with `telescope`, then $\widetilde{A}$ may contain `binocular`.

## 5 Experiments

We use proofs generated from the family of verbalized logic programs introduced in §4 to help understand how LMs reason about GSM problems. Our primary experimental manipulative is irrelevant axioms with respect to a goal theorem introduced into a logic program, which allows us to analyze how LMs fare in the face of irrelevant axioms. We generate 500 problems with varying structural, agent and entity overlap, as discussed above. See App. D.1 for more details on the make-up of the dataset. We refer to the problems *without* any irrelevant axioms as **base problems**. We additionally generate control problems that have the same number of axioms as the problems with irrelevant axioms, except that all axioms are relevant. Their shortest proofs contain the base problem's shortest proof as a subproof. This controls for the possible confounder of problem length (see, e.g., Leeb et al., 2025). We consider both non-ground queries, corresponding to questions like *"How many drones does Yanick have?"*,[13] and ground queries, corresponding to questions like *"Show that Yanick has 5 drones."*.

---

[13] Assuming numerical consistency ensures there exists at most one ground atom in the minimal Herbrand model that unifies with the non-ground query. This ensures that forward chaining would halt in finite time.

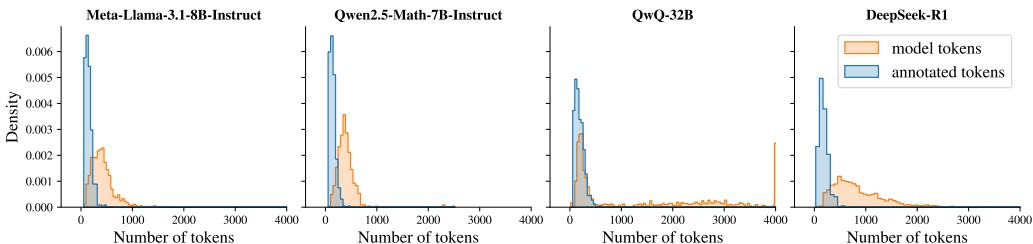

Figure 3: Number of tokens in the natural language annotation of the shortest proof as compared to model outputs for problems with three irrelevant trees ($\widetilde{A}_1 \sqcup \widetilde{A}_2 \sqcup \widetilde{A}_3$). The plot includes problems for which the model gave the correct final answer; this is why the density for annotated tokens differs between the models. We observe that LMs often use more tokens than necessary.

**Language Models.** We use one "vanilla" LM and three reasoning LMs in our experiments: Llama-3.1-8B-Instruct (Llama Team, 2024), Qwen2.5-Math-7B-Instruct (Yang et al., 2024), Qwen's reasoning model QwQ-32B, and DeepSeek-R1 (DeepSeek-AI, 2025). We generate strings using ancestral sampling, restricting the context length to $4000$ tokens.

**Prompting.** Our experimental design is based on in-context learning (Brown et al., 2020). Specifically, we use five fixed in-context examples of shortest proofs to expose the LM to proofs in our verbalized logic programs. The verbalized proofs are ordered under a DFS traversal of the theorems in the proof, such that the axioms are popped in the same order they occur in the verbalized text. See App. D.2 for the prompt.

### 5.1 Addition of Irrelevant Axioms and Answer Accuracy

We begin by analyzing how irrelevant axioms influence an LM's ability to generate the correct goal theorem for non-ground queries. We employ the two-step prompting strategy given by Kojima et al. (2022). In the first step, the model is prompted to produce a natural-language proof outlining its reasoning process. This natural-language process is then mapped to a proof in the verbalized logic program. In the second step, we prompted the LM a second time—conditioned on the proof it generated—to produce the goal theorem. In Table 2, we report model accuracy in generating the correct goal theorem. We observe that even a single irrelevant axiom reduces model performance, particularly for Llama-3.1 and Qwen2.5. Performance degrades further as additional irrelevant axioms are introduced. The performance of the most capable model, DeepSeek-R1, is nearly saturated at perfect accuracy, though slight decreases are still observed when irrelevant axioms are included. Across models, accuracy on problems containing irrelevant axioms is almost always lower than on the corresponding control examples, suggesting that irrelevance has a substantial effect on accuracy beyond what can be explained by longer problem statements. The only exception is Llama-3.1, whose performance on problems with one irrelevant axiom exceeds that of the corresponding control. QwQ-32B stands out as an outlier: for this model, performance on the control problems is significantly higher than on the original base problems. In Fig. 7 (App. D.4), we present results stratified by agent and entity overlap (§4.3). A consistent pattern emerges: such overlap between the goal and irrelevant axioms makes solving the problem more difficult. Compared to no overlap, performance drops with both kinds of overlap, suggesting both serve as heuristics during search. While drops are typically larger for agent overlap than for entity overlap, we note that this could be partially due to the entities being topically related (§4.3). In the following subsection, we analyze the proofs in greater detail to further illuminate these effects.

### 5.2 Addition of Irrelevant Axioms and Efficiency

In Fig. 3 we plot the empirical distribution over the number of tokens in the model's output and compare it to the number of tokens in the natural language annotation of the shortest proof, taking only the proofs that concluded at the correct goal theorem. This analysis, as well as those in the remainder of this section, are done on the problems with multiple irrelevant trees (i.e., the kind of structural overlap with the most axioms; §4.3). We observe that all models often use more tokens than are in the annotations, suggesting that they use more compute than necessary to prove goal theorems. This is consistent with findings on reasoning models (§2), but holds also for the Llama model. However, these results do not confirm that the models generate irrelevant theorems—they might just be more verbose than our annotations. This leads us into our efficiency analysis in line with the technical exposition in §3.2.

| | | Non-ground Queries | | | | Ground Queries | | | |
|---|---|---|---|---|---|---|---|---|---|
| | | None | Entity | Agent | Both | None | Entity | Agent | Both |
| **Llama-3.1-8B** | Efficiency | 60.2 | 43.5 | 47.5 | 34.0 | 45.3 | 34.9 | 29.9 | 23.6 |
| | Exact matches | 15 | 3 | 4 | 0 | 11 | 9 | 7 | 11 |
| | Efficiency (non-axioms) | 71.8 | 50.3 | 63.0 | 44.2 | 57.6 | 41.3 | 42.9 | 37.1 |
| | Correct Theorems | 84.5 | 82.9 | 86.0 | 85.4 | 76.0 | 77.8 | 72.1 | 69.8 |
| **Qwen2.5-Math** | Efficiency | 43.7 | 32.4 | 34.4 | 31.1 | 36.8 | 30.6 | 29.3 | 25.3 |
| | Exact matches | 8 | 0 | 1 | 0 | 10 | 1 | 2 | 1 |
| | Efficiency (non-axioms) | 57.3 | 40.1 | 51.5 | 46.2 | 50.5 | 38.1 | 41.1 | 36.0 |
| | Correct Theorems | 75.1 | 73.7 | 74.3 | 73.6 | 62.5 | 63.4 | 57.4 | 56.6 |

Table 3: We report efficiency, number of exact matches (out of 500), and efficiency restricted to non-axioms between the theorems parsed from the output generated by LMs and the shortest, most-efficient proof. Results are stratified by type of query and type of overlap between agents and entities present in the query and the irrelevant axioms. The numbers are computed based on the problems for which the model got the correct final answer. The low efficiency scores show that LMs often produce irrelevant theorems when successfully proving a goal theorem. The differences in efficiency scores across datasets of different overlaps are significant ($p < 0.001$) according to one-way ANOVA analyses. Lastly, "Correct Theorems" shows the proportion of the theorems in the shortest proof that the LM predicted.

**Efficiency Evaluation.** This analysis is performed for Llama-3.1-8B-Instruct and Qwen2.5-Math-7B-Instruct since only those models followed the required formatting (§4.2);[14] however, we perform a more crude analysis based on only the arithmetic expressions for the other two models in App. D.4. We report the efficiency metric presented in §3.2 for the problems where the LM generated the correct goal theorem, comparing against the shortest proof generated with our method (App. C).[15] We omit built-ins from the efficiency analysis since those are not verbalized. Additionally, we note that it might be the case that all irrelevant steps the models take are axioms; the LMs may simply be stating that some axioms are irrelevant to the query. We therefore also consider a metric in which only non-axiom theorems are counted. In App. D.4 we provide an additional analysis on search order.

**Efficiency for Non-ground Queries.** The main results are shown in §5.2 (non-ground queries), with scores stratified by agent and entity overlap. The efficiency scores are far from 100%, meaning that the models predict several theorems beyond the required ones present in the shortest proof. Additionally, we observe that the efficiency scores vary significantly across the type of overlap ($p < 0.001$), so we conclude that lexical information in the query has a substantial effect on proof planning. We finally comment on the results that only consider non-axioms, presented at the last row in §5.2. We again observe efficiency scores that are considerably below 100%, showing that the LMs prove theorems that are irrelevant to the query. App. D.3 gives an example where Llama-3.1-8B proves irrelevant theorems.

**Comparison to Ground Queries.** We compare the performance to the same problems when presented with ground queries. Queries tend to be non-ground in the GSM domain (Riley et al., 1983; Cobbe et al., 2021). Since LMs are heavily influenced by training data, we therefore expect them to perform better on non-ground queries. Our results on the same verbalized logic programs as before, but with ground queries, are presented on the right-hand side of §5.2. We observe lower efficiency scores, suggesting that LMs are indeed worse at proving ground theorems in this domain.

## 6 Conclusion

This paper provided a framework based on logic programming for studying deductive reasoning in language models, with a particular emphasis on efficiency. This framework enables us to disentangle efficiency due to generating irrelevant theorems from efficiency due to verbose natural language verbalizations of those theorems. We applied this framework to empirically investigate how language models perform reasoning on math word problems that have many irrelevant axioms. We found that introducing irrelevant axioms into reasoning problems leads to significantly lower answer accuracies for most models—even when controlling for problem length—and proofs that exhibit frequent detours through irrelevant theorems. Our work highlights the need to improve models in terms of reasoning efficiency, as well as the advantages to viewing *deductive* reasoning through the lens of logic programming.

---

[14]We manually verified parsing accuracy on the theorems generated by the models for a subset of 20 randomly sampled examples. An additional class representing that there is no match with any annotated theorem in the proof is included as well. The parser predicted the correct (or correctly predicted no) match in $394/397 = 99.2\%$ of the theorems for Llama-3.1-8B-Instruct, and $381/384 = 99.2\%$ of the theorems for Qwen2.5-Math-7B-Instruct.

[15]LMs often generated the same theorems multiple times. We chose to ignore such duplicates in the evaluation.

## Acknowledgments and Disclosure of Funding

We thank Juan Luis Gastaldi for useful discussion and criticisms. Andreas Opedal acknowledges funding from the Max Planck ETH Center for Learning Systems.

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

# A  Further Background

## A.1  Negation in Logic Programming

It is also useful to introduce a notion of negation into logic programming. We denote negation by $\neg$. To accommodate negation, we introduce the **extended Herbrand base** $\overline{H} \overset{\text{def}}{=} H \cup \{\neg h \mid h \in H\}$. We call an interpretation $I \subseteq \overline{H}$ **consistent** iff $b \in I \implies \neg b \notin I$. Furthermore, a logic program $\mathcal{P}$ is called **consistency-preserving** if $I$ is consistent implies $\mathrm{T}_{\mathcal{P}}(I)$ is consistent. One simple way to check whether a logic program is consistency-preserving is to inspect the program's dependency structure. Construct the **predicate dependency graph** $G$ as follows. Create a vertex $(p)$ for every distinct predicate symbol $p$ appearing in $\mathcal{P}$. For each rule $r = (b_1, ..., b_K \vdash h) \in \mathcal{P}$ and each predicate $p$ appearing among $r$'s premises, add a directed edge $(p) \to (q)$ where $q$ is $h$'s predicate symbol. Then, label the edge $-$ if $p$ is negated and $+$ otherwise. We say that $G$ has a **negative cycle** if there exists a directed cycle that contains at least one negatively labeled edge. Logic programs that do not have negative cycles are called **stratified** (Abiteboul et al., 1995, §15.2). It is easy to see that any stratified logic program has a consistency-preserving fixpoint operator.

## A.2  Forward Chaining

We give background details and pseudocode for the forward chaining algorithm; see Alg. 1. The algorithm takes a logic program $\mathcal{P} = \mathcal{R} \sqcup A$ with a set of ground goal theorems $\mathcal{H}_g$ and returns T (true) if all goal theorems in $\mathcal{H}_g$ can be proved. It uses a priority queue $Q$ and a set $\mathcal{C}$. The priority queue $Q$ is called the **agenda**. It keeps track of theorems that are to be used to prove new theorems. The order in which the axioms are pushed to $Q$ will determine the order in which they are popped and $Q$ may implement any arbitrary priority policy. For example, a last-in first-out (LIFO) policy yields a depth-first search (DFS) order, while a first-in first-out (FIFO) policy yields a breadth-first search (BFS) order. The set $\mathcal{C}$ is called a **chart**. The chart keeps track of theorems that have been proved. Theorems are added to $\mathcal{C}$ after they have been popped from $Q$.

When a theorem is popped, it is used to prove new theorems if applicable. The algorithm iterates over all rules in the program and proves all conclusions that can be proved from the popped premise together with theorems that have previously been added to $\mathcal{C}$. For a conclusion to be proved, there must exist an instantiation of the rule for which the premises in the instantiated ground rule are in $\mathcal{C}$. The algorithm iteratively removes theorems from $\mathcal{H}_g$ as they are proved, and terminates with the result T when $\mathcal{H}_g$ has been emptied. If all theorems that could be proved have been popped and $\mathcal{H}_g$ remains non-empty, the algorithm returns F (false). The algorithm may not terminate, however, if the minimal Herbrand model is infinitely large.

# B  Decidability of GSM Programs

**Proposition 1** (Decidability). *Let $\mathcal{P}_W = \mathcal{R}_W \sqcup A_W$ be a logic program where $\mathcal{R}_W$ is specified in Table 1 and $A_W$ is a numerically consistent subset of $\mathcal{R}_W$'s extended Herbrand base $\overline{H}$. Then, inference in $\mathcal{P}_W$ is decidable, i.e., it is decidable to determine whether $h \in \mathrm{T}_{\mathcal{P}}^*(A_W)$ for an arbitrary $h \in \overline{H}$.*

*Proof sketch.* First, observe that the extended Herbrand base $\overline{H}$ of $\mathcal{P}_W$ is countably infinite due to the unbounded sets of quantity constants $\mathbb{N}_x^q$ and timestamp constants $\mathbb{N}_x^t$. However, since the axioms $A_W$ were chosen to be numerically consistent and the sets $\Sigma_p^W$, $2_x^A$, and $E_x$ are all finite, it follows that $\mathrm{T}_{\mathcal{P}}^*(A_W)$—or equivalently $M$—contains only finitely many theorems for each element of $\mathbb{N}_x^t$. Consequently, reasoning in $\mathcal{P}_W$ reduces to Presburger arithmetic, as the only built-in predicate that applies to timestamps is $T_{\mathbb{N}}+1$, present in Rule (2) in Table 1. Since Presburger arithmetic is famously decidable (Presburger, 1929; Haase, 2018), it follows that inference in $\mathcal{P}_W$ is decidable. ∎

# C  Data Generation

We adapt and apply Opedal et al.'s (2025) method for generating shortest proofs with axioms that are numerically consistent. App. C.2 introduces and discusses pseudocode (Alg. 2), App. C.3 explains how we use the algorithm to generate a proof along with irrelevant axioms, and App. C.4 shows that the algorithm indeed returns a *shortest* proof. To do so, we define a class of logic programs of which $\mathcal{P}_W$ is a member, and we will present a generalization of Opedal et al.'s (2025) method

---

**Algorithm 1** Generic forward chaining

---

1: **function** FORWARDCHAINING($\mathcal{P} = \mathcal{R} \sqcup A, \mathcal{H}_g$)
2:     ▷*program $\mathcal{P} = \mathcal{R} \sqcup A$, set of ground goal theorems $\mathcal{H}_g$*
3:     $Q \leftarrow \varnothing$                                             ▷*initialize agenda*
4:     $\mathcal{C} \leftarrow \varnothing$                           ▷*initialize the chart / set of known atoms*
5:     **for** $b' \in A$:                                   ▷*push axioms to agenda*
6:        $Q.\text{PUSH}(b')$
7:     **while** $Q \neq \varnothing$:                      ▷*pop premises while available*
8:        $b' \leftarrow Q.\text{POP}()$                  ▷*pop highest priority premise*
9:        $\mathcal{C} \leftarrow \mathcal{C} \cup \{b'\}$              ▷*mark as proved by adding to chart*
10:       **for** $(b_1, ..., b_N \vdash h) \in \mathcal{R}$:      ▷*iterate over all rules in program*
11:         **if** $\exists \theta \in \Theta(\mathcal{P}) : b_1/\theta, ..., b_N/\theta \vdash h/\theta$ **and** $b_1/\theta, ..., b_N/\theta \in \mathcal{C}$:
12:           ▷*there is a new inference rule available for which all premises have been previously proved*
13:          $h' \leftarrow h/\theta$               ▷*ground conclusion to the rule*
14:          **if** $h' \notin \mathcal{C}$:               ▷*we proved a new theorem*
15:            $Q.\text{PUSH}(h')$           ▷*push to agenda*
16:           **if** $h' \in \mathcal{H}_g$:          ▷*we proved a new goal theorem*
17:             $\mathcal{H}_g \leftarrow \mathcal{H}_g \setminus \{h'\}$    ▷*mark as proved by removing from goal set*
18:             **if** $\mathcal{H}_g = \varnothing$:     ▷*the goal set is empty so we have proved all goal theorems*
19:               **return** T
20:     ▷*all goal theorems could not be proved*
21:     **return** F

---

that generates shortest proofs under any program in this class. We start by defining this class of logic programs in App. C.1 below.

## C.1   Additional Definitions

We define the **expanded Herbrand** base containing all ground and non-ground atoms—including negation (App. A.1)—as $\widetilde{H} \stackrel{\text{def}}{=} \{p(t_1, ..., t_N) \mid p \in \Sigma_p, \text{ar}(p) = N, t_1, ..., t_N \in \Sigma_x \cup \Sigma_I\} \cup \{\neg p(t_1, ..., t_N) \mid p \in \Sigma_p, \text{ar}(p) = N, t_1, ..., t_N \in \Sigma_x \cup \Sigma_I\}$. We let $C : \widetilde{H} \mapsto 2^{\Sigma_x \cup \Sigma_I}$ be a function that selects a set of terms from an input atom. We refer to $C(b)$ as the $C$-**terms** of $b$. One example of such a function in $\mathcal{R}_W$ is to return the set of agent terms from a given atom, e.g., $C(\text{cont}(A, 5, boat, 1)) = \{A\}$ and $C(\text{comp}(A, \{haruki\}, 2, boat, 1)) = \{A, \{haruki\}\}$. In the following, we will also define $C$ of an inference rule. Indeed, $r = b_1, ..., b_K \vdash h$, we define $C(r) \stackrel{\text{def}}{=} \bigcup_{k=1}^{K} C(b_k)$ as the union of the $C$-terms in the premises of the inference rule $r$. We similarly define $C$ of a hyperedge: Given a hyperedge $e = \{b_1, ..., b_K\} \rightarrowtail h$, $C(e) \stackrel{\text{def}}{=} \bigcup_{k=1}^{K} C(b_k)$.

**Definition 1.** *We say a set of inference rules $\mathcal{R}$ is $C$-**conserving** if, for every inference rule $b_1, ..., b_K \vdash h$, we have $C(b_k) \neq \{\}$ for all $k$, $C(h) \neq \{\}$, and one of the following conditions holds:*

1. *The rule is an **introduction rule** for a predicate $p$, and can be written as*

$$b_1, ..., b_K \vdash p(t_1, ..., t_N). \tag{2}$$

*Here, we require the premises to be non-$p$ atoms. We require the $C$-terms in the conclusion appear in the premises: $C(p(t_1, ..., t_N)) = \bigcup_{k=1}^{K} C(b_k)$. We also require that the $C$-terms of the premises are disjoint: $C(b_k) \cap C(b_j) = \varnothing$ for all $k \neq j$. We require that $\mathcal{R}$ have at most one introduction rule for each predicate.*

2. *The rule is an **elimination rule** for a predicate $p$, and can be written as*

$$b_1, ..., b_{K-1}, p(t_1, ..., t_N) \vdash h. \tag{3}$$

*Here, we require that the $C$-term of the conclusion is distinct from that of the premises: $b_1, ..., b_{K-1}, C(h) \cap C(b_k) = \varnothing$ for all $k = 1, ..., K - 1$. We also require that the $C$-term of the premise $p(t_1, ..., t_N)$ contains the $C$-terms of all other premises and the conclusion: $C(h) \cup \bigcup_{k=1}^{K-1} C(b_k) \subseteq C(p(t_1, ..., t_N))$. Finally, we require the $C$-terms of the other premises are disjoint: $C(b_k) \cap C(b_j) = \varnothing$ for all $0 < k, j < K$ and $k \neq j$.*

3. *The rule is a **union rule**, where there is a built-in premise requiring the $C$-terms of the conclusion be equal to the union of the $C$-terms of the premises: $C(h) = \bigcup_{k=1}^{K} C(b_k)$. This rule must be unique in $\mathcal{R}$.*

4. *The rule is an **unused rule**: it contains premises that do not unify with the conclusion of any rule.*

*Finally, if $\mathcal{R}$ has multiple elimination rules for a predicate $\mathrm{p}$, we require that those rules be distinct in the following way: For any two such rules, written $b_1, ..., b_{K-1}, \mathrm{p}(t_1, ..., t_N) \vdash h$ and $b_1', ..., b_{K-1}', \mathrm{p}(t_1', ..., t_N') \vdash h'$, for any $\theta$ such that $h/\theta = h'$, $\mathrm{p}(t_1, ..., t_N)/\theta \neq \mathrm{p}(t_1', ..., t_N')$.*

Given a $C$-conserving set of inference rules $\mathcal{R}$, we will use the following short-hand notation to refer to specific subsets of rules:

- $\mathcal{R}_{\mathrm{I}}$ is the set of introduction rules in $\mathcal{R}$.
- $\mathcal{R}_{\mathrm{E}}$ is the set of elimination rules in $\mathcal{R}$.
- $\mathcal{R}_{\mathrm{U}}$ is the set of union rules in $\mathcal{R}$.

**Definition 2.** *Let $E \colon \widetilde{H} \mapsto 2^{\Sigma_x \cup \Sigma_I}$ be a function. We say a set of inference rules $\mathcal{R}$ is $E$-**monotonic** if, for every inference rule $b_1, ..., b_K \vdash h$, we have $E(b_k) \neq \{\}$ for all $k$, $E(h) \neq \{\}$, and one of the following conditions holds:*

1. *The rule is an introduction rule; see above.*

2. *The rule is an elimination rule; see above.*

3. *The rule preserves $E$-terms: $E(b_1) = \cdots = E(b_K) = E(h)$.*

To prove the optimality of our generated proofs, we need additional constraints on the relationship between introduction and elimination rules. Let $C \colon \widetilde{H} \mapsto 2^{\Sigma_x \cup \Sigma_I}$ be a function. Consider each elimination rule $b_1, ..., b_{K-1}, \mathrm{p}(t_1, ..., t_N) \vdash h$, and let $C_b \overset{\text{def}}{=} \bigcup_{k=1}^{K-1} C(b_k)$ and $C_h \overset{\text{def}}{=} C(h)$. Construct a hypergraph using the following procedure: Start by mapping the atom $\mathrm{p}(t_1, ..., t_N)$ to a vertex $v_0$, i.e., we set $\ell(v_0) = (\mathrm{p}(t_1, ..., t_N))$. Repeat the following: For any vertex $v$ that contains $C$-terms in both $C_b$ and $C_h$ (i.e., $C(\ell^{-1}(v)) \cap C_b \neq \varnothing$ and $C(\ell^{-1}(v)) \cap C_h \neq \varnothing$), and for any introduction rule $b_1', ..., b_K' \vdash h' \in \mathcal{R}$, add a vertex for each unified premise $b_k/\theta$ where $\ell(h'/\theta) = v$ and a hyperedge from $\{\ell^{-1}(b_k/\theta)\}$ to $\ell^{-1}(v)$. We say the elimination rule is $C$-**locally reducible** if either: (1) the hypergraph contains no edges, or (2) there exists a vertex in this hypergraph that identifies with the conclusion of the elimination rule $h$. If all elimination rules in $\mathcal{R}$ are locally reducible, we say $\mathcal{R}$ is **in harmony**. One consequence of harmony is that, in any proof $\mathcal{P}$ containing a proof step with an elimination rule, if the subproof of the premise $\mathrm{p}(t_1, ..., t_N)$ does not contain any axioms with $C$-terms from both $C_b$ and $C_h$, which is true of any proof generated by the procedure in App. C, then the conclusion of the elimination rule must be an axiom of the subproof. Therefore, $\mathcal{P}$ is not optimal.

We note that $\mathcal{R}_{\mathrm{W}}$ is **agent-conserving**, i.e., $\mathcal{R}_{\mathrm{W}}$ is $C$-conserving where $C$ is the function that returns the agents of any atom. In particular, if we inspect the rules in Table 1, we observe that (1a) and (1b) are elimination rules for comp, and (1c) is the corresponding introduction rule. Rules (2a) and (2b) are elimination rules for transfer. Rule (4) is an elimination rule for rate. There are no corresponding introduction rules for transfer and rate. (5a) is an elimination rule for compeq and (5b) is the corresponding introduction rule. Rule (3) is a union rule. Rule (6) is an unused rule. $\mathcal{R}_{\mathrm{W}}$ is also **entity-monotonic** with (4) being the elimination rule for rate. $\mathcal{R}_{\mathrm{W}}$ is also in harmony, with respect to both agents and entities: (1a) and (1b) are each locally reducible with (1c). (5a) is locally reducible with a combination of (5b) and (1c).

A **ground** substitution $\{(X_m, t_m)\}_{m=1}^{M}$ is a typed substitution where $t_m \in \Sigma_x^k$, for all $m \in [M]$. We define $\Theta_{\mathrm{G}}(\mathcal{P})$ to be the set of all ground substitutions under the logic program $\mathcal{P}$.

## C.2 Applying Opedal et al.'s (2025) Method

Opedal et al. (2025) provide a method to generate the shortest proof for a goal $h_g$, i.e., the shortest $(\ell^{-1}(A_{\mathrm{W}}), \ell^{-1}(h_g))$-hyperpath that exists in a proof forest over the Herbrand base; see §3.2. We present the pseudocode in Alg. 2. The method takes any logic program $\mathcal{P}$ with $C$-conserving rules, e.g., $\mathcal{P}_{\mathrm{W}}$ as described in Table 1, a goal theorem $h_g$, a set of forbidden ground theorems $A_\times$, and a

**Algorithm 2**

1. **function** SAMPLESHORTESTPROOF($\mathcal{P}, h_g, A_\times, D$)
2.     ▷*logic program $\mathcal{P}$ (e.g., Table 1), goal theorem $h_g$, forbidden ground theorems $A_\times$, max depth $D$*
3.     let $\mathcal{R}$ be the inference rules in $\mathcal{P}$
4.     $\mathscr{P} \leftarrow \varepsilon$                                                       ▷*initialize an empty hyperpath*
5.     QUEUE $\leftarrow [\,]$                                ▷*initialize queue for conclusions to expand and their depth*
6.     QUEUE.PUSH$((0, h_g))$                               ▷*$h_g$ is distance $0$ from $h_g$*
7.     SET $\leftarrow \{h_g\} \cup A_\times$
8.     **while** QUEUE $\neq \varnothing$:
9.        $(d, h) \leftarrow$ QUEUE.POP$()$
10.       **if** $d > D$:                               ▷*exit if stopping criterion has been met*
11.          **break**
12.       ▷*first, sample the inference rule for the next proof step*
13.       $\mathcal{R}' \leftarrow \{r \mid r = (b_1', ..., b_K' \vdash h') \in \mathcal{R}_\text{I} \cup \mathcal{R}_\text{E} \cup \mathcal{R}_\text{U}, \exists \theta \in \Theta_\text{G}(\mathcal{P}) : h'/\theta = h\}$
14.       $r = (b_1', ..., b_K' \vdash h') \sim$ SAMPLE$(\mathcal{R}')$
15.       ▷*next, we sample the substitution*
16.       $\Theta' \leftarrow \{\theta \mid \theta \in \Theta_\text{G}(\mathcal{P}), h'/\theta = h, b_k'/\theta \notin \text{SET}\}$
17.       ▷*sample novel and distinct $C$-terms whenever possible*
18.       let $C_\text{SET} \overset{\text{def}}{=} \bigcup_{x \in \text{SET}} C(x)$
19.       $\Theta' \leftarrow \Theta' \cap \{\theta \mid |\bigcup_{k=1}^K C(b_k') \setminus C(h')| = |\bigcup_{k=1}^K C(b_k'/\theta) \setminus C_\text{SET}|\}$
20.       **if** $r \in \mathcal{R}_\text{U}$:                            ▷*sample widest possible union rules*
21.          $\Theta' \leftarrow \Theta' \cap \{\theta \mid C(b_k'/\theta) \cap C(b_j'/\theta) = \varnothing \text{ for } k \neq j\}$
22.       $\theta \sim$ SAMPLE$(\Theta')$
23.       **for** $k = 1$ **up to** $K$:
24.          SET $\leftarrow$ SET $\cup \{b_k'/\theta\}$
25.          **if** $r \in \mathcal{R}_\text{E}$ **and** $k \neq K$:           ▷*the last premise of elimination rules are axioms*
26.             QUEUE.PUSH$((d + 1, b_k'/\theta))$
27.       ▷*construct the hyperedge for this new proof step, and prepend it to the hyperpath*
28.       $\mathscr{P} \leftarrow (\{b_1, ..., b_K\} \rightarrowtail h) \circ \mathscr{P}$
29.     **return** $\mathscr{P}$

maximum depth $D$. The set $A_\times$ becomes relevant when generating irrelevant axioms, which will be discussed in App. C.3. For now, we assume that $A_\times = \varnothing$. The algorithm will terminate and return a proof of depth $D$. In our experiments with $\mathcal{P}_\text{W}$, we use the following arguments: We sample the goal $h_g$ to be a ground atom of the form cont($A$, $Q$, $E$, $T$)$/\theta$ for some $\theta \in \Theta(\mathcal{P}_\text{W})$. For the agent set we make an arbitrary choice, sampling an agent set with cardinality $\{1, 2, 3, 4\}$ with probabilities $\{1 \mapsto \text{\textonehalf}, 2 \mapsto \text{\textonesixth}, 3 \mapsto \text{\textonesixth}, 4 \mapsto \text{\textonesixth}\}$. We restrict the cardinality to $4$ in order avoid generating GSM problems that are too large; see App. D.1 for dataset statistics. The timestamp is set to an arbitrary value larger than the value of $D$, which ensures that the timestamps remain in $\mathbb{N}_x^t$ throughout the generation procedure. We sample $D$ uniformly at random from $\{1, 2, 3\}$ for every generated proof.

Sampling proceeds recursively in a top-down manner. We then sample an inference rule $b_1, ..., b_K \vdash h$, i.e., where $h$ is the conclusion. This yields a tree. We then repeat the procedure recursively for each leaf node until the stopping criterion, i.e., required depth, has been reached. Importantly, all premises are sampled *without* replacement; this ensures that any individual theorem will not be used more than once in the generated proof, i.e., the proof is acyclic. For example, suppose we have cont($\{haruki\}$, $5$, $boat$, $1$) and sample an instantiation of Rule (1a) in Table 1. We generate two new premises, cont($\{abu\}$, $3$, $boat$, $1$) and comp($\{haruki\}$, $\{abu\}$, $2$, $boat$, $1$), where the agent $\{abu\}$ is a new agent that does not appear elsewhere in the proof. In Alg. 2, this is handled by maintaining a set SET of generated non-ground atoms, and by restricting the substitutions such that new $C$-terms are mapped to new objects. We only sample atoms that are not in SET.

We comment on two more details of Alg. 2. First, whenever we sample an elimination rule $b_1, ..., b_{K-1}, \text{p}(t_1, ..., t_N) \vdash h$, we require the last premise, $\text{p}(t_1, ..., t_N)$, to be an axiom, i.e., we do not recursively apply the algorithm on this premise. This is due to the fact that the rules in $\mathcal{P}$ are in harmony, and so the addition of introduction rules preceding elimination rules could result in locally-reducible regions in the output proof, and so the proof would not be shortest. Second, if we

sample a union rule, we always require the $C$-terms of the premises to be singleton sets. Otherwise, a shorter proof could have been obtained by several instantiations of the union rule.[16]

Throughout the algorithm, we sample substitutions from $\Theta_G(\mathcal{P})$. By substituting under the constraints laid out by the arithmetic built-ins we ensure numerical consistency. However, numerical substitutions may fail due to unsatisfiable constraints, i.e., $\Theta'$ in Alg. 2 may become empty before we sample from it. We therefore perform rejection sampling until a successful substitution has been found. In a few cases, this may run prohibitively long. We therefore retry a maximum of 1000 times before rejecting the candidate proof and sampling a new one.

### C.3 Generating Math Word Programs with Irrelevant Axioms

We apply Alg. 2 described in the previous subsection to generate irrelevant axioms. As mentioned in §4.3, we augment a logic program $\mathcal{P}_W = \mathcal{R}_W \sqcup A_W$—as sampled by the method described above—with $M$ additional sets of irrelevant axioms $\widetilde{A}_1 \sqcup \cdots \sqcup \widetilde{A}_M$. In simple terms, we do so by applying the sampling procedure again, once for every $\widetilde{A}_m$, with additional restrictions to not generate duplicate axioms. As we will show later, this procedure for generating irrelevant axioms requires the additional constraint that the inference rules $\mathcal{R}$ in the logic program $\mathcal{P}$ be $E$-monotonic. With each application of the sampling procedure, we provide a perturbed goal argument $h_g$. At least one $C$-term or $E$-term of the original goal will be replaced with a new value (e.g., in $\mathcal{P}_W$, either the agent, the entity, or both will be perturbed). Precisely *which* term is perturbed depends on the experimental setting (see §4.3). The set of forbidden theorems $A_\times$ is initialized to be the theorems in the shortest proof (including the axioms) from $\mathcal{P}_W = \mathcal{R}_W \sqcup A_W$, thus, avoiding generating those theorems. The same is done incrementally for the irrelevant axioms as they are generated, guaranteeing disjoint sets.

Moreover, for practical reasons, the distribution from which we sample is slightly modified. In particular, we restrict cont predicates to have singleton agent sets; otherwise, the number of irrelevant axioms could get very large. Furthermore, we exclude the rate predicate, as we found that overlapping agents and entities could sometimes be hard to instantiate with a rate atom, since it requires a particular semantic relationship between the two entity terms in the atom, e.g., *"apples"* per *"basket"*. Enforcing these restrictions led to substantially more efficient rejection sampling during the substitution part of Alg. 2.

When ordering the axioms in natural language, it is undesirable to have a predictable ordering of relevant and irrelevant axioms, e.g., appending all irrelevant axioms to follow the relevant ones. We therefore randomly reorder them in a manner that respects the ordering of the values of the timestamp.

### C.4 Uniqueness and Optimality of Generated Proofs

Here, we will show that the above procedure for generating proofs produces a shortest proof—a proof from axioms $A$ to a goal $h_g$ for which there does not exist a shorter proof of $h_g$ from $A$. Furthermore, we will show that each generated proof is *unique*: There are no other shortest proofs with the same axioms and goal theorem.

$C$-conservation, monotonicity, and harmony of the inference rules alone is insufficient to guarantee that the proofs generated by the procedure described in App. C are shortest. We need additional properties of the generated proofs, as well as a few additional definitions.

A **linear chain** is a sequence of hyperedges $e_1, ..., e_M$ where the head of every edge is in the tail of the next edge (except for the last edge). More precisely, for all $m$, we have $e_m = \{b_{m,1}, ..., b_{m,K}\} \rightarrowtail h_m$, and for all $0 < m < M$, $h_m \in \{b_{m+1,1}, b_{m+1,2}, ...\}$.

Given an $(A, h_g)$-proof $\mathcal{P} = ((V, E), \ell)$, and an atom $x \in \overline{H}$ in the extended Herbrand base $\overline{H}$ (App. A.1) where $\ell^{-1}(x) \in V$, a **subproof** of $x$ is an $(A, x)$-proof $\mathcal{P}_x = ((V_x, E_x), \ell)$ where $V_x \subseteq V$ and $E_x \subseteq E$. Using this, we can define the subproof relation: If $\mathcal{P}_x$ is a subproof of some atom in $\mathcal{P}$, we write $\mathcal{P}_x \preceq \mathcal{P}$. We say a subproof $\mathcal{P}_x$ of $x$ *contains* an atom $y$, or similarly, $y$ *is in* $\mathcal{P}_x$, if there is an $(A, x)$-hyperpath where $\ell^{-1}(y)$ is in the head or tail of any edge in the hyperpath. Similarly, we say a subproof $\mathcal{P}_x$ of $x$ *contains* a $C$-term $t$, or $t$ *is in* $\mathcal{P}_x$, if there is some $y$ such that $\mathcal{P}_x$ contains $y$ and $t \in C(y)$.

---

[16]This is analogous to folding and unfolding (Tamaki & Sato, 1984).

**Theorem 1** (Generated proofs are shortest and unique). *Let $\mathcal{R}$ be a set of inference rules that are $C$-conserving, $E$-monotonic, and in harmony,[17] let $\mathcal{P}$ be an $(A, h_g)$-proof in $\mathcal{R} \sqcup A$ generated by the procedure in App. C.3, and let $\widetilde{A}$ be the set of irrelevant axioms generated by App. C.3. There does not exist an $(A', h_g)$-proof $\mathcal{P}'$ in $\mathcal{R} \sqcup A \sqcup \widetilde{A}$ such that $|\mathcal{P}'| \leq |\mathcal{P}|$.*

To prove Thm. 1 we introduce two lemmas. First, in Lemma 1, we show that the shortest proofs (without irrelevant axioms) are unique. That is, for a given goal theorem and set of axioms, there is no shortest proof that is different than the one we generate. Then, we show that the irrelevant axioms generated in App. C.3 generated through Alg. 2 can not be used to yield another shortest proof of the goal theorem; this is done in Lemma 2. Taken together, these lemmas imply Thm. 1.

**Lemma 1.** *Let $\mathcal{R}$ be a $C$-conserving set of inference rules, and let $A$ be a set of axioms. Then, for all theorems $h_g$ in $\mathcal{R} \sqcup A$ such that $(A, h_g)$-proof $\mathcal{P}$ could be generated by Alg. 2, we have that $\mathcal{P}$ is $h_g$'s unique shortest proof in $\mathcal{R} \sqcup A$.*

*Proof.* We perform structural induction on the subproof relation $\preceq$.

We first observe that $\mathcal{P}$ can not have cycles: By definition, a cycle contains several vertices labeled with the same theorem. This is not possible since Alg. 2 maintains a set of already sampled theorems SET that can never be sampled again: Specifically, line 16 excludes theorems in SET from being sampled and line 24 adds generated theorems to SET.

**Base Case.** The base case is the smallest possible proof containing a single axiom that is also the goal theorem, i.e., an $(A, h_g)$-proof $\mathcal{P}$ of $h_g$ with $|\mathcal{P}| = 1$. This is clearly the unique shortest proof in this case.

**Inductive Case.** Consider the last hyperedge $\{\ell^{-1}(b_1), ..., \ell^{-1}(b_K)\} \rightarrowtail \ell^{-1}(h)$, i.e., the hyperedge for which the head is labeled with the goal theorem $h$ of the subproof. The inductive hypothesis states that for any atom $x \neq h$ in $\mathcal{P}$, the subproof of $x$ in $\mathcal{P}$ is the unique shortest $(A, x)$-proof in $\mathcal{R} \sqcup A$. We consider the various cases of inference rules in the logic program for the last hyperedge and show that the unique shortest proof of the conclusion is obtained by combining the unique shortest proofs of the premises under the inference rule corresponding to that hyperedge.

We now consider four cases.

*Case 1: (Introduction).* The inference rule corresponding to the last hyperedge is an instantiation of an introduction rule:
$$b_1, ..., b_K \vdash \mathrm{p}(t_1, ..., t_N). \tag{4}$$
Because there are no other introduction rules for $\mathrm{p}$, $C(\mathrm{p}(t_1, ..., t_N)) = \bigcup_{k=1}^{K} C(b_k)$, and the atoms in the subproofs of the premises are disjoint, the above introduction rule is the only way to prove $\mathrm{p}(t_1, ..., t_N)$. Thus, we conclude that $\mathcal{P}$ is the unique shortest proof of $\mathrm{p}(t_1, ..., t_N)$.

*Case 2: (Elimination).* The inference rule corresponding to the last hyperedge is an instantiation of an elimination rule: $b_1, ..., b_{K-1}, \mathrm{p}(t_1, ..., t_N) \vdash h$. Due to the restriction in the generative process—see App. C.2—the premise $\mathrm{p}(t_1, ..., t_N)$ is not expanded when generating $\mathcal{P}$ because it is an axiom. Furthermore, since $C(h) \cap C(b_k) = \varnothing$ for all premises $b_k$, it is impossible to prove $h$ using only the atoms in the subproofs of the premises $b_1, ..., b_{K-1}$. In fact, since $C(h) \subseteq C(\mathrm{p}(t_1, ..., t_N))$, the axiom $\mathrm{p}(t_1, ..., t_N)$ *must* appear in any proof of $h$. The logic program may have other elimination rules for the same predicate $\mathrm{p}$, but since such rules are distinct due to $C$-conservation, those other rules cannot be used to derive $h$ from $b_1, ..., b_{K-1}, \mathrm{p}(t_1, ..., t_N)$. Therefore, we conclude that $\mathcal{P}$ is the unique shortest proof of $h$.

*Case 3: (Union).* The inference rule corresponding to the last hyperedge is an instantiation of a union rule: $b_1, ..., b_K \vdash h$. Since the generation procedure in App. C.2 only generates hyperedges corresponding to union rules where the premises have disjoint $C$-terms, i.e., $C(b_k) \cap C(b_j) = \varnothing$ for all $k \neq j$, and there is no way to prove $h$ other than the union rule, this case is identical to *Case 1* above.

*Case 4: (Unused).* The inference rule corresponding to the last hyperedge is an instantiation of an unused rule. But the procedure in App. C.2 never includes such inference rules, so this is impossible.

∎

---

[17]With respect to both $C$ and $E$.

**Lemma 2.** *Let $\mathcal{R}$ be a set of inference rules that is $C$-conserving, $E$-monotonic, and in harmony, let $(A, h_g)$-proof $\mathcal{P}$ be the proof in $\mathcal{R} \sqcup A$ generated by App. C.3, and let $\widetilde{A}$ be the set of irrelevant axioms generated by App. C.3. Then, there does not exist an $(A', h_g)$-proof $\mathcal{P}'$ in $\mathcal{R} \sqcup A \sqcup \widetilde{A}$ such that $A' \neq A$ and $|\mathcal{P}'| \leq |\mathcal{P}|$.*

*Proof.* We offer a proof by contradiction. By way of contradiction, assume there exists an $(A', h_g)$-proof $\mathcal{P}'$ in $\mathcal{R} \sqcup A \sqcup \widetilde{A}$ such that $A' \neq A$ and $|\mathcal{P}'| \leq |\mathcal{P}|$. Take $\mathcal{P}'$ to be the shortest such proof. Because, by construction of the algorithm, the proof $\mathcal{P}$ consists of a single hyperpath starting from $A$. Thus, $A' \neq A$ implies there exists an $a \in A'$ where $a \notin A$. Recall that $\mathcal{P} = ((V, E), \ell)$ is itself a proof forest. Let $C(\mathcal{P}) \overset{\text{def}}{=} \bigcup_{e \in E} C(e)$ be the set of $C$-terms in $\mathcal{P}$, i.e., the **relevant $C$-terms**. Similarly, let $E(\mathcal{P}) \overset{\text{def}}{=} \bigcup_{e \in E} E(e)$ be set of $E$-terms in $\mathcal{P}$, i.e., the relevant $E$-terms. We now consider two cases.

*Case 1: $C(a) \subseteq C(\mathcal{P})$ (The irrelevant axiom only has relevant $C$-terms).* The generation procedure described in App. C.3 can only generate an irrelevant atom with relevant $C$-terms when generating proof trees that have $E$-terms that are distinct from those in the relevant proof. Therefore, the $E$-terms in $\mathcal{P}'$, i.e., the irrelevant $E$-terms, are disjoint from those in $\mathcal{P}$, i.e., the relevant $E$-terms. Take the linear chain $e_1, ..., e_M \in \mathcal{P}'$ such that $a$ is in the tail of $e_1$ and $h_g$ is the head of $e_M$. Let $h_m$ be the head of edge $e_m$. Select the first $e_m$ such that $E(h_m)$ contains an $E$-term that is not relevant and $E(h_{m+1})$ has only relevant $E$-terms. There must be at least one such edge since $h_g$ has only relevant $E$-terms but $a$ does not. Because $\mathcal{R}$ is $E$-monotonic, only elimination rules allow an $E$-term from a premise to be absent from the conclusion, and so $e_{m+1}$ must be an instantiation of an elimination rule

$$b_1, ..., b_{K-1}, \mathrm{p}(t_1, ..., t_N) \vdash h, \tag{5}$$

and $E(h) \cup E(b_k) = E(\mathrm{p}(t_1, ..., t_N))$, for any $k$. The $E$-terms of $h_m$ and $h_{m+1}$ are included in $E(\mathrm{p}(t_1, ..., t_N))$. Thus, the premise $\mathrm{p}(t_1, ..., t_N)$ must contain both relevant and irrelevant $E$-terms. Note that in the generation procedure described in App. C.3, no axiom is generated that contains both relevant and irrelevant $E$-terms. Therefore, $\mathrm{p}(t_1, ..., t_N)$ must be derived from axioms other than $a$. However, because $\mathcal{R}$ is in harmony, the subproof of this premise must contain $h$, and so the proof $\mathcal{P}'$ is not shortest, which is a contradiction.

*Case 2 $C(a) \nsubseteq C(\mathcal{P})$ (The irrelevant axiom has an irrelevant $C$-term).* Take the linear chain $e_1, ..., e_M \in \mathcal{P}'$ such that $a$ is in the tail of $e_1$ and $h_g$ is the head of $e_M$. Let $h_m$ be the head of edge $e_m$. Select the first $e_m$ such that $C(h_m) \nsubseteq C(\mathcal{P})$ and $C(h_{m+1}) \subseteq C(\mathcal{P})$. There must be at least one such edge since $h_g$ has only relevant $C$-terms but $a$ does not. Since the $\mathcal{R}$ is $C$-conserving, only elimination rules allow a $C$-term from a premise to be absent from the conclusion, and so $e_{m+1}$ must be an instantiation of an elimination rule

$$b_1, ..., b_{K-1}, \mathrm{p}(t_1, ..., t_N) \vdash h, \tag{6}$$

and $C(h) \cup \bigcup_{j=1}^{K-1} C(b_j) \subseteq C(\mathrm{p}(t_1, ..., t_N))$. That is, the $C$-terms of $h_m$ and $h_{m+1}$ are included in $C(\mathrm{p}(t_1, ..., t_N))$. Thus, the premise $\mathrm{p}(t_1, ..., t_N)$ must contain both relevant and irrelevant $C$-terms. Note that in the generation procedure described in App. C.3, no axiom is generated that contains both relevant and irrelevant $C$-terms. Therefore, $\mathrm{p}(t_1, ..., t_N)$ must be derived from other axioms. However, since $\mathcal{R}$ is in harmony, the subproof of this premise must contain $h$, and so the proof $\mathcal{P}'$ is not shortest, which is a contradiction. ∎

We can now prove Thm. 1 using the above lemmas.

**Theorem 1** (Generated proofs are shortest and unique)**.** *Let $\mathcal{R}$ be a set of inference rules that are $C$-conserving, $E$-monotonic, and in harmony,[18] let $\mathcal{P}$ be an $(A, h_g)$-proof in $\mathcal{R} \sqcup A$ generated by the procedure in App. C.3, and let $\widetilde{A}$ be the set of irrelevant axioms generated by App. C.3. There does not exist an $(A', h_g)$-proof $\mathcal{P}'$ in $\mathcal{R} \sqcup A \sqcup \widetilde{A}$ such that $|\mathcal{P}'| \leq |\mathcal{P}|$.*

*Proof.* By Lemma 1, there exists one unique shortest proof $\mathcal{P}$ of a goal theorem from a set of relevant axioms and this is the proof we generate as the ground-truth proof. By Lemma 2, there will be no additional proofs $\mathcal{P}'$ where $|\mathcal{P}'| \leq |\mathcal{P}|$ when generating irrelevant axioms through our procedure. Therefore, all our generated proof systems have a unique shortest proof, which is identical to the ground-truth proof that is generated through our procedure. ∎

---

[18]With respect to both $C$ and $E$.

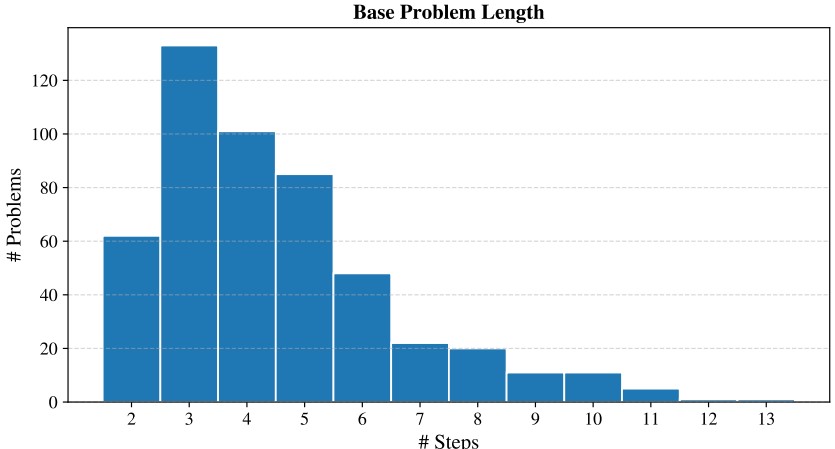

Figure 4: Histogram of the number of hyperedges (i.e., inference rule applications) present in the shortest proof for the GSM programs in our dataset.

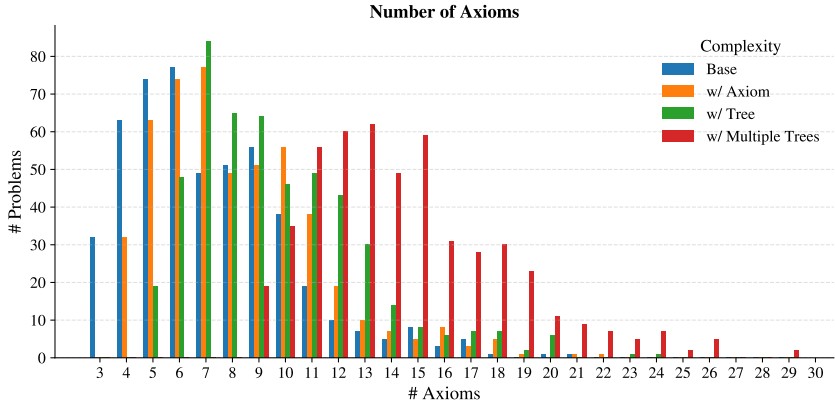

Figure 5: Histogram of the total number of axioms, including irrelevant ones, per type of complexity of the GSM programs in our dataset.

# D  More Details on Experiments

## D.1  Dataset Statistics

Here, we present more statistics on the datasets used in our experiments (§5). Fig. 4 shows the distribution of the number of edges in the proofs in the base problems, which is equivalent to the number of non-axiom theorems. The numbers range from 2 to 13.[19] The distribution is skewed towards shorter problems. This was done by design in order to prevent saturation when evaluating models with different levels of capabilities.

Fig. 5 shows the number of axioms the GSM problems contain, including irrelevant ones. As expected, adding one irrelevant axiom (w/ axiom) yields a distribution that is shifted by exactly one additional axiom as compared to the base distribution. When adding a single irrelevant tree (w/ tree), the distribution shifts further and spreads out. This trend is further amplified when adding multiple irrelevant trees (w/ multiple trees).

## D.2  Prompt

See Fig. 6 for the prompt used in our experiments.

---

[19]For reference, GSM8K (Cobbe et al., 2021) includes problems with 2 to 8 steps.

Figure 6: Two-stage prompting method used in our experiments. We first prompt the model with in-context examples (5-shot) and the test problem. After generation, we re-prompt the model to generate the final numerical answer.

### D.3 Example Proof

In this section we give an example proof in natural language. Consider the following problem, with the irrelevant axioms marked in **bold**:

> **Anatola has 67 keys. Ichabod has 23 packets. Clayborne has 55 crayons.** Pieter has 20 parrots. **Ichabod has 30 packets less than Winnah.** Pieter then gives Yorgos 6 parrots. **Clayborne has 96 crayons less than Imogene. Anatola has 55 keys less than Rahel.** Dennie has 19 parrots. Dianna owns 4 parrots. The number of parrots that Edgar has more than Pieter is the same as the difference between the number of parrots that Dennie has compared to Dianna. Edgar has 3 parrots more than Eddie. How many parrots does Eddie have in all?

Note that the only axioms that are relevant are the ones containing information about parrots. The shortest proof for this problem requires the application of four inference rules and can be written in natural language as follows:

1. Pieter has 20 parrots.
2. Pieter then gives Yorgos 6 parrots.
3. So Pieter has 20 - 6 = 14 parrots.
4. Dennie has 19 parrots.
5. Dianna owns 4 parrots.

6. So the difference between the number of parrots Dennie and Dianna have is 19 - 4 = 15.
7. The number of parrots that Edgar has more than Pieter is the same as the difference between the number of parrots that Dennie has compared to Dianna.
8. So Edgar has 14 + 15 = 29 parrots.
9. Edgar has 3 parrots more than Eddie.
10. So Eddie has 29 - 3 = 26 parrots.

In this case, employing a best-first search heuristic based on lexical overlap with the query (i.e., prioritizing theorems that contain information about parrots) will yield the above shortest proof. Llama-3.1-8B-Instruct did not opt for that strategy. Instead, it generated the following, inefficient solution to this problem:

1. Anatola has 67 keys.
2. Anatola has 55 keys less than Rahel.
3. So Rahel has 67 + 55 = 122 keys.
4. Ichabod has 30 packets less than Winnah.
5. So Winnah has 23 + 30 = 53 packets.
6. Clayborne has 96 crayons less than Imogene.
7. So Imogene has 55 + 96 = 151 crayons.
8. Pieter has 20 parrots.
9. Pieter then gives Yorgos 6 parrots.
10. So Pieter has 20 - 6 = 14 parrots.
11. The number of parrots that Edgar has more than Pieter is the same as the difference between the number of parrots that Dennie has compared to Dianna.
12. Dennie has 19 parrots and Dianna has 4 parrots.
13. The difference between Dennie and Dianna is 19 - 4 = 15.
14. So Edgar has 15 more parrots than Pieter.
15. Edgar has 15 + 14 = 29 parrots.
16. Edgar has 3 parrots more than Eddie.
17. So Eddie has 29 - 3 = 26 parrots.
Therefore, the answer (arabic numerals) is 26.

### D.4 Further Empirical Results

**Accuracy Stratified by Agent and Entity Overlap.** Fig. 7 shows further stratified results that were mentioned in §5.1. We briefly note that these results suggest that the models might make use of a heuristic based on such overlap. (Recall that the problem sets are otherwise identical.) One could reasonably expect that a model could learn such a heuristic, based on previous work suggesting that models may derive the conclusion to a proof step greedily within a forward pass before initiating the step in the output tokens (Kudo et al., 2024; Wu et al., 2024).

**Parsing Reasoning Model Outputs.** Since QwQ-32B and DeepSeek-R1 did not generate output corresponding to strings in our verbalized program, we performed a separate, more crude analysis for those two models. Rather than matching the full generated token sequence, we match only the arithmetic expressions. They are matched with the ground-truth expressions from the built-ins of the corresponding inference rules. This can only be done for theorems that are not axioms, since only those require an arithmetic built-in (i.e, arithmetic expression) to prove. Arithmetic expressions are extracted flexibly, accounting for formatting irregularities and natural language that may be interleaved. The parser may also predict no match for a particular model output if no matches are found.

We manually verified parsing accuracy (macro-average) on model outputs over 20 randomly selected example problems. The parser predicted the correct match (or correctly predicted no match) in 94.4% of cases for QwQ-32B and 94.8% of cases for DeepSeek-R1.

Table 4 displays the results. We observe results that are overall consistent with the main conclusions of the paper; even state-of-the-art models like DeepSeek-R1—despite performing well in terms of accuracy (Table 2)—appear to generate theorems that are irrelevant to the goal theorem. As with the two models discussed in the main text, the efficiency scores decrease when there is agent and/or entity overlap between the irrelevant axioms and the goal theorem.

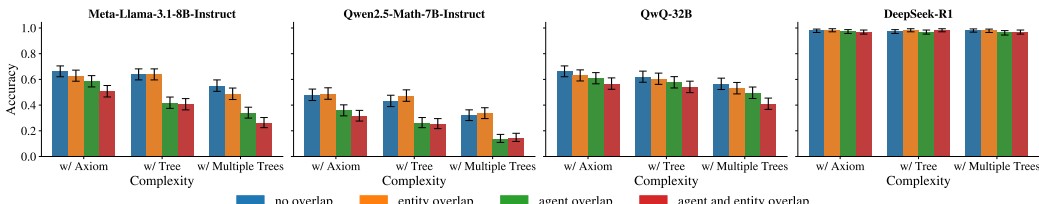

Figure 7: Results on problems with irrelevant axioms from Table 2 as average answer accuracy (%) when stratified by different kinds of lexical overlap between the irrelevant axioms and the query. The error bars show the 95% confidence interval using Wilson (1927) score intervals.

| | | **Non-ground Queries** | | | |
| | | None | Entity | Agent | Both |
|---|---|---|---|---|---|
| **QwQ-32B** | Efficiency (non-axioms) | 77.1 | 63.8 | 72.2 | 63.1 |
| **DeepSeek-R1** | Efficiency (non-axioms) | 89.4 | 67.7 | 77.3 | 67.8 |

Table 4: Additional efficiency analysis for QwQ-32B and DeepSeek-R1, using a more crude parser that only extracts arithmetic expressions corresponding to the arithmetic built-ins in Table 1. The efficiency score can therefore only be presented for theorems that are not axioms. Results are stratified and computed in the same way as §5.2. The differences in efficiency scores across datasets of different lexical overlaps are significant ($p < 0.001$) according to one-way ANOVA analyses.

**Search Order.** While our results in §5.2 suggest that LMs make use of information in the query to prove goals, they also reveal that LMs generate irrelevant theorems. Thus, LMs appear to use some heuristic, albeit an imperfect one, in their reasoning. Here, we present a limited analysis on whether this heuristic could be in combination with either DFS or BFS. We do so by examining the order in which intermediate theorems, both relevant and irrelevant, are generated by the LM, and comparing this order to the respective reference orders. Furthermore, we compare the LM's ordering to the theorems in the shortest proof as visited in DFS order as a control.

Before presenting the results, we make note of a few limitations of this analysis. First, the only irrelevant theorems we consider in the DFS and BFS orderings are those that are generated by Alg. 2 when sampling the axioms for the irrelevant goal theorems $\widetilde{h}_1...\widetilde{h}_M$ (§4.3). In addition, we note that the in-context examples were ordered according to DFS; see §5. However, the examples were chosen so that the ordering of theorems that are not axioms would be the same under both DFS and BFS. We therefore ignore the axioms as we compare the orderings.

Concretely, the DFS and BFS orderings correspond to the order in which atoms are popped from the chart $\mathcal{C}$ in Alg. 1, where: (i) $\mathcal{C}$ is implemented as a stack for DFS and a queue for BFS, (ii) the axioms are popped from the agenda $Q$ in the order in which they are presented (i.e., pushed) to the model in natural language (as described in §5), and (iii) we do not include pops of axioms.

As our metric, we compute the Levenshtein distance between the sequence of indices produced by the LM and the ground truth reference ordering under the different search orders, normalized by the longest of the two sequences. The results are shown in Fig. 8. We observe that the models' search orders are closer to DFS than to BFS across all types of query overlap. Furthermore, for Llama-3.1, they are closer to the DFS-based ordering required for the shortest proof than DFS when there is no agent or entity overlap between the goal and the irrelevant axioms. Qwen2.5-Math appears to be less efficient however, consistent with the results in §5.2. These findings suggest that LMs tend to follow a depth-first exploration of the proof space in combination with the superficial heuristic. However, future work should perform a more rigorous analysis to confirm these preliminary findings.

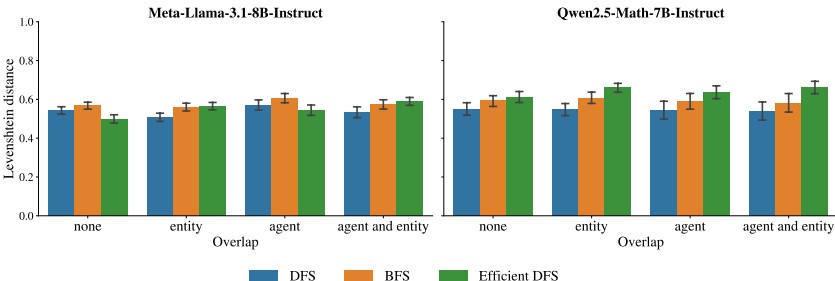

Figure 8: Average Levenshtein distance between Llama-3.1-8B-Instruct's search order and depth-first search (DFS), breadth-first search (BFS), and an efficient DFS which only visits relevant theorems. The plot is based on problems for which the model gave the correct answer. The Levenshtein distances are normalized by the length of the longest string to take values in $[0, 1]$; lower values mean shorter distances.

