# OpenReview forum: "Are Language Models Efficient Reasoners? A Perspective from Logic Programming"
_NeurIPS.cc/2025/Conference — NeurIPS 2025 poster_

### Official Review · Reviewer_XHVw · 2025-07-03

**Clarity:** 3
**Significance:** 3
**Originality:** 3
**Rating:** 5
**Confidence:** 4

**Summary:**

The paper studies a very basic question: "Is it true that LLM performance degrades when irrelevant facts are provided". In particular, the paper investigates the question whether the reasoning gets worse (unnecessary steps are performed).

In order to do this, the paper uses word problems since smallest proofs for them are unique (this allows measuring non-optimality of the proof generated by LLM). It compares proofs generated by three open-source models with different amount of irrelevant facts and showed that with the amount of irrelevant information increasing the performance is decreasing.

**Questions:**

1. It would be good to add variance to the plots on Figure 1.
2. line 231: It would be good to explain why smallest proof is a tree-like proof; for many proof systems the difference between tree- and proof-like proofs is big.
3. line 164: it would be good to mention algorithms such as A* here.

**Ethical Concerns:**

["NO or VERY MINOR ethics concerns only"]

**Final Justification:**

I reviewed the response from the authors and other concerns from the other reviewers, and my opinion hasn't changed:
while I agree that the example is somewhat artificial, proving that at least this settign LLM "think" as we expect them to think is important step in better understanding LLM capabilities.

**Limitations:**

yes

**Paper Formatting Concerns:**

No concerns

**Quality:**

3

**Strengths And Weaknesses:**

The paper is interesting and written well; however, the answer that the paper shows is expected and the paper doesn't give any guidance on how this result could be applied in practice. In addition, the problem studied is very far from real-life examples.

---

> ### Author Rebuttal · Authors · 2025-07-30
>
> Thank you for the overall positive review and helpful feedback.
>
> We can definitely include some discussion on practical implications of the results. While we did not spell this out in the paper, our findings that LMs are generally unable to distinguish relevant from irrelevant information signifies that there is room for improvement in this regard in the training paradigm of current LMs. For instance, efficiency might be improved by penalizing inefficient proofs through the reward function used for finetuning LMs with RL techniques and this could be studied in future work. We will add a discussion to the conclusion.
>
> In response to your questions:
> 1. This is a great suggestion; we will make sure to include it in the final version of the paper.
> 2. The smallest proof is always tree-shaped because it does not contain cycles, all vertices are connected, and each premise is used at most once (by construction). We will add this clarification to the paper.
> 3. We actually mentioned A* in a previous draft, but decided to omit it since, in our setting, the costs of the hyperedges (or proof steps) are uniform. But we agree that it is a good idea to add it back in for the sake of contextualization in the final version of the paper, and will do so.

---

> > ### Comment · Reviewer_XHVw · 2025-08-05
> >
> > I am perhaps missing something, but why cannot you find a shorter proof where some premises are used more than once?

---

> > > ### Author Response · Authors · 2025-08-05
> > >
> > > We just meant to say that we do not generate normal-form proofs where a premise is used more than once. This is guaranteed in our generative process by sampling new premises for each inference rule.
> > >
> > > In general, however, you are right that a most efficient (normal form) proof could have premises that are used more than once and may therefore have (undirected) cycles. For example, consider the proof of commutativity of conjunction:
> > >
> > > 1. A & B by Axiom
> > > 2. B by &-elimination[1]
> > > 3. A by &-elimination [1]
> > > 4. B & A by &-introduction [2,3]
> > >
> > > In the above proof, the axiom is used twice.

---

> > > > ### Comment · Reviewer_XHVw · 2025-08-05
> > > >
> > > > Oh, I got it! Thanks!

---

### Official Review · Reviewer_WAXh · 2025-07-03

**Clarity:** 2
**Significance:** 2
**Originality:** 3
**Rating:** 4
**Confidence:** 3

**Summary:**

This paper explores the efficiency of large language models (LLMs) when solving arithmetic reasoning problems, particularly when faced with irrelevant information. The authors introduce a formal proof system for grade-school math problems, allowing for precise definitions of efficiency and irrelevance through the concept of a "normal form" for proofs. Their experiments reveal that LLMs' answer accuracy significantly drops when problems include irrelevant facts, even if seemingly related, and that these models often generate inefficient proofs by taking unnecessary steps. The analysis suggests that LLMs tend to employ a depth-first search strategy, partially guided by lexical overlap with the query, but this heuristic isn't always effective in filtering out noise.

**Questions:**

1.  Clarify the "Distraction" Conclusion for Advanced Models:
    *   Question/Suggestion: The paper's conclusion states that irrelevant axioms lead to "significantly lower answer accuracies". However, for the most capable model, DeepSeek-R1, Table 2 indicates performance is "nearly saturated" with only "slight decreases" in accuracy. While Figure 2 shows LMs, including DeepSeek-R1, use more tokens (indicating efficiency loss), this may be an expected behavior for complex models. Could the authors refine their conclusion to clearly differentiate between the impact on *accuracy* vs. *efficiency*, particularly for advanced reasoning models, and provide a more nuanced discussion of what "distraction" entails in this context for state-of-the-art LLMs?
    *   Criteria for Score Change: A revised or more detailed analysis that precisely delineates how irrelevance affects accuracy versus efficiency across different model capabilities would significantly enhance the paper's clarity and the empirical robustness of its claims, positively impacting the evaluation of its significance and quality.

2.  Distinguish Novelty from Prior Work (e.g., GSM-Logic/GSM-NoOp):
    *   Question/Suggestion: The paper states it "extends their method to generate proofs with irrelevant axioms" and creates "problems with many irrelevant axioms". Please explicitly clarify the significant differences and novelties of your irrelevant fact generation methodology and proof evaluation approach compared to existing benchmarks that incorporate irrelevant information or provide programmatic/proof-based evaluation, specifically addressing works such as "GSM-Logic" (e.g., GSM-NoOp), which might also specify programs as proofs. Does your generation of irrelevant axioms from other *normal-form proofs* and control over "complexity" and "lexical overlap" offer distinct advantages?
    *   Criteria for Score Change: A clear articulation of the methodological advancements and empirical distinctions from relevant prior work (like GSM-Logic's approach to irrelevant information or structured proofs) would greatly enhance the paper's perceived originality and contribution, leading to a higher evaluation of its significance.

3.  Formalize "Complexity" and "Lexical Overlap" Definitions:
    *   Question/Suggestion: While the paper provides a robust theoretical framework for proof systems, the "Complexity" (Axiom, Tree, Multiple Trees) and "Lexical overlap" (entity, agent) properties used to generate irrelevant facts are described conceptually. To align with the paper's formal emphasis, could the authors provide more formal definitions or cite specific references for these terms and their quantification?
    *   Criteria for Score Change: Introducing more formal definitions or appropriate academic references for "Complexity" and "Lexical Overlap" would enhance the paper's rigor and clarity in its experimental design, improving its overall quality.

4.  Provide a Complete Natural Language Problem Example:
    *   Question/Suggestion: The paper describes generating natural language annotations for problems with irrelevant axioms but does not show a complete example of such an augmented problem (i.e., the actual input given to the LLM). Table 1 only provides templates for inference rules. To help readers intuitively assess task complexity and the reasonableness of the generated questions, could the authors include at least one full natural language example of a problem augmented with interleaved irrelevant axioms?
    *   Criteria for Score Change: Presenting a concrete, end-to-end natural language example of an augmented problem would significantly improve the clarity and practical understanding of the experimental setup, positively impacting the evaluation of the paper's clarity and quality.

5.  Discuss Scalability of "Most Efficient Proof" for Harder Reasoning:
    *   Question/Suggestion: The paper's evaluation relies on comparing LM outputs to a "provably unique normal form" of the "most efficient ground-truth proof". While this is a strength for the arithmetic proof system studied due to its confluence, the paper also notes that "Proving all possible facts... is computationally intractable in general". Could the authors discuss the generalizability and scalability of this "most efficient proof" concept and evaluation methodology to more complex or open-ended reasoning problems where a unique normal form might not exist, finding it is intractable, or where trial-and-error is an inherent and normal part of the problem-solving process? How might this framework adapt or be limited in such scenarios?
    *   Criteria for Score Change: A thoughtful discussion acknowledging the limitations of the "most efficient proof" concept for more challenging and open-ended reasoning domains, and exploring how their methodology might need to be adapted or its claims re-contextualized for such scenarios, would strengthen the paper's overall significance and provide a more comprehensive assessment of its applicability.

**Ethical Concerns:**

["NO or VERY MINOR ethics concerns only"]

**Final Justification:**

The author's response has addressed my concerns about the formality of some definitions and the significance of the work, though I am not sure if the revised version will fully reflect my suggestions and concerns. Anyway, I would like to raise my score.

**Limitations:**

yes

**Quality:**

3

**Strengths And Weaknesses:**

### Strengths

*   Strong Theoretical Framing and Formalization: The paper establishes a well-defined theoretical framework for reasoning and introducing irrelevant axioms. It formally defines a proof system as a 3-tuple (L, A, R) and provides precise characterizations of concepts like efficiency and irrelevance. This formalization, including the property of confluence that ensures a unique, most efficient normal form for proofs, is a key strength that allows for rigorous evaluation.
*   Novel Evaluation Methodology: The study introduces a new dataset specifically designed to evaluate Language Models (LMs) in the presence of many irrelevant facts, a setting that more closely mirrors real-world reasoning challenges than typical evaluation paradigms. The method for generating problems with irrelevant axioms, sourced from other normal-form proofs and controlling for complexity and lexical overlap, is a new perspective to me.
*   Comprehensive Proof Verification: Unlike many studies that only verify the final answer, this paper evaluates the full proof generated by the LM, comparing it to the most efficient ground-truth proof. This is done via an automatic evaluation based on semantic parsing, which is a more fine-grained approach to assessing reasoning.

### Weaknesses

*   Limited Impact of Irrelevance on Advanced Models' Accuracy: While the paper concludes that irrelevant axioms lead to significantly lower answer accuracies, this finding appears less pronounced for the most capable models like DeepSeek-R1, where performance is "nearly saturated" with only "slight decreases". The observation that LMs use more test-time compute than necessary might be an expected behavior for complex models, and similar overthinking has been noted in other models. This slightly weakens the claim that these LMs are easily "distracted" in terms of *accuracy*, though their *efficiency* is clearly impacted.
*   Informal Definitions for Problem Generation Properties: Despite a strong theoretical framework for proof systems, the definitions of "complexity" (Axiom, Tree, Multiple Trees) and "lexical overlap" (entity, agent) in the irrelevant fact generation process are described conceptually rather than with the same level of formal rigor. More formal definitions or references would enhance the clarity and robustness of these experimental parameters.
*   Lack of Concrete Natural Language Examples: The paper describes the generation of natural language annotations for problems and proofs. However, it does not provide full natural language examples of the augmented problems given to the LMs. This makes it difficult for readers to intuitively grasp the actual complexity and reasonableness of the tasks from an LM's perspective (Table 1 shows only templates).
*   Potential Scalability and Generalizability Concerns of "Most Efficient Proof": The reliance on a "provably unique normal form" for the "most efficient ground-truth proof" is a strength for the specific arithmetic proof system studied. However, this concept of a single "most efficient proof" might not generalize well to more complex or open-ended reasoning problems where multiple valid proofs exist, or where trial-and-error is an inherent part of the problem-solving process, and an "efficient" path is not always guaranteed or even discoverable upfront.

---

> ### Author Rebuttal · Authors · 2025-07-30
>
> Thank you for the thorough review and for the structured, concrete feedback. We are happy that you appreciate the formal treatment of proof systems and the novelty and comprehensiveness of the evaluation methodology. We have incorporated most of the feedback, which definitely strengthens the paper, and respond to each of your questions below.
>
> 1. Clarify the "Distraction" Conclusion for Advanced Models:
> * Thanks for noticing this. We will weaken the claim about irrelevance leading to significantly lower accuracies in the conclusion and throughout the paper, since there is only a small, nonsignificant effect for DeepSeek-R1. Regarding accuracy vs. efficiency: As you allude to, it seems like more capable models such as DeepSeek-R1 are not affected much in terms of accuracy, but they still use more tokens than they do for the corresponding base problems (i.e., those without irrelevant axioms). We created a new plot to show that this is indeed the case, which we will add to the appendix and discuss in 5.2. Thus, our findings suggest that more capable models are “distracted” only in the sense that it takes them more tokens to find the answer, but they still do, generally, find the correct answer. These findings are complementary to recent findings on “overthinking” of reasoning models. This was a helpful suggestion; we will add more discussion in the paper.
>
> 2. Distinguish Novelty from Prior Work (e.g., GSM-Logic/GSM-NoOp):
> * We agree that we can improve the clarity of the methodological contributions as compared to prior work, in particular in relation to Mirzadeh et al. (2025) and Shi et al. (2023). The main innovation in our work as compared to theirs is in our method that generates problems with irrelevance *automatically*, whereas these papers seem to have added irrelevant statements manually. In addition, while they augmented problems from existing datasets, we created new problems from scratch. Our method thus avoids a potential bias from memorizing the efficient solution seen during training. In addition, we note the following differences to GSM-Symbolic’s No-Op dataset: (i) they seem to only include *one* irrelevant statement, whereas we may have multiple that could be simultaneously included as premises in the same proof; (ii) they describe irrelevance in an informal manner so the semantics of these irrelevant statements are unclear; (iii) it is hard to know further details about their setup since the details  in the paper are scarce and the dataset is still not available (there is currently an open issue about this in their github repo); we will release our datasets and generation code with the final version of the paper.
>
> * We will incorporate the above discussion into the Related Work section.
>
> 3. Formalize "Complexity" and "Lexical Overlap" Definitions:
> * Thanks, this is a good point. We will revise that section to be more formal and precise. As for the two specific terms you mention, we propose the following: “Lexical overlap” between two atomic formulas can be defined as the type(s) of atom(s), if any, for which the same atom value is present in both atomic formulas. “Complexity” of the irrelevance is about hypergraph connectivity; an irrelevant axiom is one disconnected vertex in the proof, a tree is a component with multiple vertices, and “Multiple Trees” refers to multiple mutually disconnected components, each having multiple vertices. (Recall that a component is a connected subgraph that is not part of any larger connected subgraph.)
>
> 4. Provide a Complete Natural Language Problem Example:
> * This is a great suggestion.  We have actually already added (1) a figure in the introduction which illustrates a proof and a corresponding natural language annotation as well as (2) an example problem in natural language with the ground-truth normal-form proof and a model-generated proof to a new Appendix C.
>
> 5. Discuss Scalability of "Most Efficient Proof" for Harder Reasoning:
> * It is true that the exact type of analysis we propose is restricted to settings with confluent proof systems where there exists a unique most efficient proof. With that said, many of the notions we discuss do actually carry over to more general settings. We are not sure what you mean by “open-ended reasoning” but if we think of it as a non-confluent proof system where there exist several normal forms, we could evaluate efficiency of solutions to such problems simply by the merit of being in (or close to) *some* normal form (i.e., a non-reducible proof). Moreover, we could still compare the number of proof steps in the different normal forms and we may be specifically interested in the proof that has the fewest, which could be checked. Indeed, math word problems may not have unique normal forms in general and we agree that it is important to make note of this limitation.
>
> * In cases where there does not exist any normal form proof at all, it is not possible to adapt our evaluation method. In such cases you might have to resort to more crude metrics such as the number of repeated/redundant proof steps or the number of generated tokens.
>
> * Finally, we note that our statement from the introduction that proving all possible facts could be intractable is not contradictory to there being a unique, most-efficient normal form. This statement is about proving *all* facts implied by the proof system, i.e., not only those relevant to a specific query.
>
> * Thank you for raising these points; we have added a limitations section to the end of the paper and will incorporate the above discussion into that section.

---

> ### Comment · Reviewer_WAXh · 2025-08-06
>
> Thank you for the authors' response. I now have a better understanding of the technical details of the paper. I would increase my score if no other severe issues raised by subsequent discussions.

---

> > ### Author Response · Authors · 2025-08-08
> >
> > Thanks again for the helpful feedback that helped us improve the paper.

---

### Official Review · Reviewer_6aPK · 2025-07-03

**Clarity:** 3
**Significance:** 2
**Originality:** 3
**Rating:** 4
**Confidence:** 3

**Summary:**

The work "Language Models Are Inefficient Reasoners" investigates how current large language models handle reasoning in the presence of many irrelevant axioms within grade‑school arithmetic word problems. The paper argues that language models are inefficient reasoners when tasks contain many distractor facts, with the example of achieving normal form efficiency in formal proof systems. The authors adapt the GSM arithmetic generator to insert irrelevant axioms while preserving consistency, varying the complexity (single axiom / single tree / three trees) and lexical overlap (none/entity/agent/both).

The paper finds that the answer accuracy falls sharply with irrelevance (e.g. Llama‑3 drops from 65% to 41% when three irrelevant trees are added). Parsed proof steps reveal high recall but low precision, indicating many superfluous inferences, while search‑order distance analysis shows traces are closer to DFS than BFS. The authors conclude that LMs employ a depth‑first style strategy guided by a weak lexical heuristic.

**Questions:**

1. Significance: can you add statistical tests (confidence intervals or McNemar/paired bootstrap tests) for accuracy and precision to support claims of significant degradation?
2. Model diversity: can you include at least one closed‑weight frontier model (e.g. GPT‑4o) via API to check whether scaling or RLHF reduces inefficiency?
3. Semantic parser reliability: please provide an error analysis of the 5–6% unmatched proof steps; could systematic misses from the parser distort the results?

**Ethical Concerns:**

["NO or VERY MINOR ethics concerns only"]

**Final Justification:**

The responses were adequate but did not raise my evaluation which is already at "borderline accept".

**Limitations:**

Mostly adequately addressed in the paper. Some additional points:
* reliance on template parsing restricts coverage;
* proofs are limited to depth ≤ 3.

**Quality:**

3

**Strengths And Weaknesses:**

Strengths:
* [*originality*] as far as I know, this is the first systematic study of irrelevant‑fact noise on LM proof search; the work introduces a novel confluence‑based efficiency metric;
* [*quality*] the formal proofs are correct, and dataset construction algorithms are clear;
* [*structure*] the paper is easy to read and understand, and illustrative examples are helpful;
* [*significance*] dataset and metrics may be introduced into standard benchmarking for researchers working on reasoning, search and CoT evaluation; the paper also promises to release the code for data generation which is always helpful.

Weaknesses:
* [*originality*] the work deals with arithmetic domain only and relies on an existing GSM generator;
* [*quality*] there are only four models in the comparison, and 500 samples per condition may be insufficient;
* [*quality*] the semantic parser is imperfect (94-95% accuracy) which could bias precision/recall;
* [*significance*] the work is limited to short arithmetic proofs, so it is unclear whether the conclusions transfer to other domains (e.g., logical or commonsense reasoning).

---

> ### Author Rebuttal · Authors · 2025-07-30
>
> Thank you for the review and useful feedback, and for acclaiming the paper for its novel metric, correctness, and clarity. We respond to your questions below and have made several improvements to the paper based on your feedback.
>
> > Significance: can you add statistical tests (confidence intervals or McNemar/paired bootstrap tests) for accuracy and precision to support claims of significant degradation?
>
> Thanks, this is a great suggestion. We performed one-way ANOVA tests for the precision, recall and "precision for non-axioms" scores and found statistically significant F-scores (p < 0.001) for precision and precision for non-axioms, both for data with ground and nonground queries. Thus, we conclude that the means of these scores are different across the types of lexical overlap between irrelevant facts and the query, supporting our claim on degradation in terms of efficiency with more lexical overlap (as measured by precision). We will add this analysis to the paper.
>
> > Model diversity: can you include at least one closed‑weight frontier model (e.g. GPT‑4o) via API to check whether scaling or RLHF reduces inefficiency?
>
> We wanted to use open-weight models in this study in order to ensure replicability, improve understanding of the results in relation to, e.g., architectural choices and training, and overall promote a culture of open science. Our frontier model of choice is DeepSeek-R1, which turns out to be less efficient in terms of tokens generated than Llama-3.1-8B-Instruct and Qwen2.5-Math-7B-Instruct (Figure 2). With that said, if you still feel that including a model like GPT-4o to the analysis is worthwhile, we will be happy to do so even if it means paying OpenAI for API access.
>
> > Semantic parser reliability: please provide an error analysis of the 5–6% unmatched proof steps; could systematic misses from the parser distort the results?
>
> Thanks, this was a useful suggestion. We performed an analysis on the unmatched proof steps and found that most of them were on container-type predicates (i.e., statements about possession), especially for the Llama model. The majority of them were misclassified because the model gave the correct intermediate numerical answer without the arithmetic equation that implies that answer. We incorporated such cases into the parser which boosted the parsing accuracy to 99.2% for the Llama model and 97.7% for the Qwen model.
>
> > [originality] the work deals with arithmetic domain only and relies on an existing GSM generator;
>
> While it is true that we rely on an existing GSM generator (Opedal et al., 2025), we would like to add that adapting their method to our study required some nontrivial extensions. For instance, we need to ensure that the irrelevant axioms can be used in inference rules with each other but not with the relevant ones, which is non-trivial. We have detailed some of these innovations in 4.2 and App. B, but we will make sure to better clarify these technical nuances in the final version.
>
> > [significance] the work is limited to short arithmetic proofs
>
> To clarify, the base problems (i.e., excluding irrelevant axioms) may have up to 12 proof steps, i.e., a lot more than 3. (Note that depth is different from the number of proof steps.) Compare, for instance, to GSM8K (Cobbe et al., 2021), which includes problems with 2-8 proof steps. Moreover, the dataset can easily be extended to even longer problems if desired in future studies. To make this clearer, we created a histogram over the number of proof steps which we will add to the appendix and reference in 4.2. We will also make sure to improve the clarity regarding the data distribution more generally in that section.
>
> Lastly, we would like to thank the reviewer for mentioning additional limitations. We have included a Limitations section at the end of the paper and will add these points in the discussion there.

---

> > ### Comment · Reviewer_6aPK · 2025-08-08
> >
> > Thank you for addressing my concerns! I remain with my opinion that this paper is not a breakthrough but an adequate and interesting contribution that can be accepted.

---

### Official Review · Reviewer_Bqyz · 2025-07-08

**Clarity:** 2
**Significance:** 3
**Originality:** 2
**Rating:** 4
**Confidence:** 3

**Summary:**

The paper evaluates how efficiently language models perform arithmetic proof search when faced with irrelevant information. Using a formal proof system and uniquely defined efficient proofs, the authors show that even strong LMs often include unnecessary steps, especially when irrelevant facts overlap with the query. The models tend to follow depth-first-like search guided by imperfect heuristics, revealing a gap between getting correct answers and reasoning efficiently.

**Questions:**

- In Table 2, for Llama-3.1-8B under “w/ Axiom”, why is the accuracy with irrelevant axioms (60.8%) higher than the control (54.0%)? Does this suggest that irrelevant information improves performance, or is it a typo?
- Could you clarify the results in Table 3? It’s unclear how the experiments demonstrate that LMs are empirically worse at proving ground theorems.

**Ethical Concerns:**

["NO or VERY MINOR ethics concerns only"]

**Final Justification:**

Most of concerns have been addressed except for that I am still a bit worried the applicability of the theoretical framework to more complex problems.

**Limitations:**

yes

**Paper Formatting Concerns:**

n.a.

**Quality:**

2

**Strengths And Weaknesses:**

Strengths
- Well-controlled experiments using a clearly defined proof system.

Weaknesses
- While the controlled setup is appreciated, it may be overly simplistic and may not generalise well to real-world reasoning tasks.
- The negative impact of irrelevant premises is a known issue, even outside rigorously defined proof systems, which may limit the novelty of the conclusions.
- The claim that LLM proof search follows a DFS-like ordering is interesting, but the experimental evidence supporting it is not entirely convincing.

---

> ### Author Rebuttal · Authors · 2025-07-30
>
> Thank you for your review and feedback. We respond to your questions and reservations below.
>
> > While the controlled setup is appreciated, it may be overly simplistic and may not generalise well to real-world reasoning tasks.
>
> Thanks for raising this point. We believe there is a natural trade-off between higher ecological validity on the one hand and a more principled/formal understanding on the other; we chose to prioritize the latter in this study. Besides, our approach has additional advantages such as mitigating data leakage, and the problems, while synthetic, are reminiscent of  those found in popular datasets such as GSM8K (Cobbe et al., 2021). We take your point and will rewrite the introduction to better motivate our study.
>
>
> > The negative impact of irrelevant premises is a known issue, even outside rigorously defined proof systems, which may limit the novelty of the conclusions.
>
> We agree that previous papers (Shi et al., 2023,  Mirzadeh et al., 2025) have reported that irrelevant information has a negative effect on LM reasoning. Note that we make several contributions beyond these papers: (i) while previous papers analyzed problems with *one* irrelevant premise, often placed at the same position in the text, we provide more complex problems where there are irrelevant premises at various positions in text that can be used in further deduction; (ii) our study considers not only answer accuracy, but also proof efficiency; (iii) we have a method to automatically generate problems with irrelevant premises of various kinds, which is more scalable; (iv) we formally characterize the notion of irrelevance, which enables a more principled understanding of the findings. Thus, our study can be viewed as refining and extending findings from previous papers, which we acknowledge in the paper, e.g., at l85-7 and and l300-1. With that said, we agree that we can do a better job at this and will incorporate the above discussion into the related work section. Moreover, we would be grateful for any pointers to other related literature we may have missed that made similar contributions!
>
>
> > The claim that LLM proof search follows a DFS-like ordering is interesting, but the experimental evidence supporting it is not entirely convincing.
>
> We are pleased that the reviewer also sees the value in studying search order. We do indeed find that the LM’s search order is closer to DFS than to BFS, but, for the sake of clarity, we would like to add that we additionally find evidence that LMs use a heuristic based on lexical overlap with the query, which yields a more efficient algorithm than basic DFS. That is, we did not intend to claim that they follow DFS and will revise any claims that can be perceived that way.
>
> > In Table 2, for Llama-3.1-8B under “w/ Axiom”, why is the accuracy with irrelevant axioms (60.8%) higher than the control (54.0%)? Does this suggest that irrelevant information improves performance, or is it a typo?
>
> Thanks for this question; these numbers deserve a note in section 5.1. It does *not* suggest that irrelevant information improves performance, since the performance on the dataset with irrelevance is still lower than the performance on the base problems (see leftmost column). (As a reminder, the control problems have the same number of axioms as the corresponding problems with irrelevance, except with all axioms being relevant.) It appears to be an outlier; for all other models and datasets, the control accuracy is higher than the non-control one. We did not find a clear explanation for these numbers after manually looking at some of the model responses. We will clarify this point in the paper.
>
> > Could you clarify the results in Table 3? It’s unclear how the experiments demonstrate that LMs are empirically worse at proving ground theorems.
>
> Precision gives the size of the overlap between the LM's reasoning steps and the steps in the efficient ground truth divided by the number of LM steps, while recall gives the overlap divided by the number of steps in the efficient ground truth. Precision therefore gives a score for efficiency, and recall gives a score for correctness in the steps taken. We observe generally lower scores for both precision and recall when the LMs are given ground queries. Therefore, we conclude that they are less efficient and less correct for ground queries as compared to nonground queries. We will spell this out more clearly in the paper.

---

### Decision · Program_Chairs · 2025-09-17

**Decision:**

Accept (poster)

**Comment:**

The paper Language Models Are Inefficient Reasoners: An Analysis on Arithmetic Proof Search investigates whether large language models (LMs) can reason efficiently when faced with irrelevant information in arithmetic proof tasks. It introduces a formal proof system with unique normal forms to define proof efficiency, generates problems with irrelevant axioms of varying complexity and lexical overlap, and evaluates several LMs (Llama-3.1-8B, Qwen2.5-Math-7B, QwQ-32B, DeepSeek-R1). The key findings are that irrelevant information reduces accuracy (especially for smaller models), models often include superfluous proof steps even when correct, and their search strategy resembles depth-first search guided by weak lexical heuristics.

### Strengths of the paper:

- It provides a rigorous formalization of proof systems, irrelevance, and efficiency, including a confluence property guaranteeing a unique efficient proof.

- It introduces a scalable dataset with irrelevant axioms (from other proofs), controlling for complexity and lexical overlap, going beyond prior single-noise setups.

- It goes beyond final answer checking, comparing full LM proofs against efficient ground truth using semantic parsing.

- It shows that LMs use more test-time compute than necessary, often proving irrelevant facts, with clear patterns of depth-first-like search.

- Paper is well structured, examples are provided, and code/dataset release is promised.

### Weaknesses of the paper:

- Experiments focus only on short arithmetic proofs; generalization to broader reasoning tasks remains unclear.

- It relies on a limited set of models (no closed-weight frontier models like GPT-4o) and ~500 examples per condition.

- Semantic parser has ~94–95% accuracy, raising small risks of bias in precision/recall estimates.

- Core finding (irrelevant info harms LMs) is expected and has been noted in prior works; contributions are more incremental than groundbreaking.

- Framework’s assumption of unique most-efficient proofs may not extend to domains where multiple valid proofs exist.

### Primary reasons for Accept (Poster)

The paper is technically solid and makes a careful, principled contribution by formalizing proof efficiency and irrelevance in LM reasoning. While not a major breakthrough, it advances the evaluation of LMs by introducing proof-level metrics and a systematic dataset that stress-tests efficiency under distraction. Its findings about DFS-like search and heuristic use are insightful for understanding LM reasoning. The limitations in domain scope and novelty prevent a spotlight, but the work is strong enough for poster acceptance.

### Summary of the discussion and rebuttal

During rebuttal, the authors addressed reviewers’ concerns by 1) Differentiated their automatic irrelevant-axiom generation from prior manual/no-op approaches, emphasizing scalability and avoidance of data leakage. 2) Added statistical tests (ANOVA, p < 0.001) for degradation claims, more formal definitions of “complexity” and “lexical overlap,” and example augmented problems. 3) Conducted error analysis, boosting parsing accuracy to >97% by incorporating systematic misses. 4) Weakened statements about accuracy drops for advanced models like DeepSeek-R1, instead framing the issue as inefficiency (more tokens used). 5) Added discussion on the restricted applicability of unique-proof evaluation and how methods could adapt to non-confluent systems. Overall, the rebuttal strengthened the paper’s clarity, empirical support, and positioning relative to prior work, leaving consensus around a Poster acceptance.